

# Assimilation of GPM-retrieved Ocean Surface Meteorology Data for Two Snowstorm Events during ICE-POP 2018

Xuanli Li[1], Jason B. Roberts[2], Jayanthi Srikishen[3], Jonathan L. Case[4], Walter A. Petersen[2], GyuWon Lee[5], and Christopher R. Hain[2]

[1]University of Alabama in Huntsville, Huntsville, Alabama, USA

[2]NASA Marshall Space Flight Center, Huntsville, Alabama, USA

[3]Universities Space Research Association, Huntsville, Alabama, USA

[4]ENSCO, Inc./NASA SPoRT Center, Huntsville, Alabama, USA

[5]Kyungpook National University, Daegu, Republic of Korea

*Correspondence to: Xuanli Li (xuanli@nsstc.uah.edu)*

*For publication in Geoscientific Model Development*

**Abstract.** As a component of the National Aeronautics and Space Administration (NASA) Weather Focus Area and GPM Ground Validation participation in the International Collaborative Experiments for PyeongChang 2018 Olympic and Paralympic Winter Games (ICE-POP 2018) field research and forecast demonstration programs, hourly ocean surface meteorology properties were retrieved from the Global Precipitation Measurement (GPM) microwave observations for January – March 2018. In this study, the

retrieved ocean surface meteorological products – 2-m temperature, 2-m specific humidity, and 10-m wind speed were assimilated into a regional numerical weather prediction (NWP) framework to explore the application of these observations for two heavy snowfall events during the ICE-POP 2018: 27-28 February, and 7-8 March 2018. The Weather Research and Forecasting (WRF) model and the community Gridpoint Statistical Interpolation (GSI) were used to conduct high resolution simulations and data assimilation

experiments. The results indicate that the data assimilation has a large influence on surface thermodynamic and wind fields in the model initial condition for both events. With cycled data assimilation, positive influence of the retrieved surface observation was found for the March case with improved quantitative precipitation forecast and reduced error in temperature forecast. A slightly smaller yet positive impact was also found in the forecast of the February case.



## 1. Introduction

Cold season storms make great contributions to the global water cycle and influence local water supplies for household, agriculture, and manufacturing uses. In addition, winter sports tourism involving outdoor activities such as skiing, snowboarding, snowmobiling, ice fishing, etc., is a large market segment for mid to high latitude regions. On the other hand, hazardous winter weather including blizzards, ice storms, freezing rains, and heavy snow, often disrupt transportation, affect outdoor activities, cause delays and closures of airports, government offices, schools, and businesses, produce widespread and extensive property damages, losses of electricity, and present hazards to human health and even loss of life (Changnon, 2003, 2007; Call, 2010; FEMA, 2016; NCDC, 2016).

Accurate and timely forecast of the onset, duration, intensity, type, and spatial extent of precipitation is a major challenge in winter weather forecasting (Garvert et al., 2005; Ralph et al., 2005, 2010; Novak and Colle, 2012). These are especially important factors for providing support to ensure the success of highly weather-sensitive venues, such as the winter Olympic and Paralympic games that were held in South Korea during February and March 2018. Many factors can contribute to the development of winter precipitation, including synoptic forcing (e.g., warm advection, differential vorticity advection), strong baroclinicity in the presence of moisture sources (e.g., near coastlines), and large-scale environmental instability in the warm sector of a mid-latitude cyclone. For regions that contain complex terrain in proximity to large bodies of water (such as the Korean peninsula), local circulations and seas air-sea interactions also play important roles in determining the phase and amount of precipitation (Niziol et al., 1995; Kain et al., 2000; Schultz et al., 2002; O'Hara et al., 2009; Alcott and Steenburgh, 2010; Novak and Colle, 2012; Schuur et al., 2012; Novak et al., 2014; Roller et al., 2016).

In Korean peninsula, the weather and climate regime during the winter months is largely driven by the seasonal reversal of winds across eastern Asia and the western North Pacific Ocean from predominantly south/south-westerlies during the boreal summer months, to north/north-easterlies during boreal winter (Chang et al. 2006). The east Asian winter monsoon (EAWM) months are considered between November and March, and largely drive the temperature and precipitation patterns across Korea. The dominant weather features associated with the EAWM consist of a strong low pressure in the Aleutian region of Alaska, a cold-core Siberian-Mongolian High, and low-level northeasterly winds along the Russian east coast. Variability in the strength of the EAWM (described in Zhang et al. 1997) has been correlated to El Niño/Southern Oscillation phase (where La Niña [El Niño] corresponds to stronger [weaker] EAWM), and snowpack anomalies during the Autumn/Winter across Siberia, eastern Russia, and northeastern China (positive snowpack anomalies lead to stronger EAWM). A stronger EAWM corresponds to strong Aleutian lows, Siberian-Mongolian highs, a stronger subtropical jet stream across eastern Asia, and deeper troughs in eastern





Asia (Chang et al. 2006). Lee et al. (2010) found that, contrary to expectations, storm track activity is reduced during stronger EAWM and increased during weaker EAWM years.

Due to the prevailing EAWM regime, the Korean peninsula can feel the effect of severe winter weather in the form of rapidly-deepening mid-latitude cyclones and occasional cold surges from the Siberian-Mongolian semi-permanent high. Bomb cyclogenesis is most common along the Japanese coastline, but

because of the Korean peninsula's proximity to the Yellow Sea (west) and Sea of Japan (east), strong baroclinicity can develop between the cold continental polar air over land and the warmer waters that provide abundant fluxes of heat and moisture into the atmosphere. Therefore, rapid deepening of cyclones can also occur in the vicinity of the Korean peninsula. In their satellite-era climatology of east Asian extratropical cyclones, Lee et al. (2020) showed that the Korean peninsula feels the influence of extratropical cyclones

originating in three preferred regions: Mongolia, East China, and the Kuroshio current along the southern/eastern coast of Japan. Using reanalysis data back to 1958, Zhang et al. (2012) found similar results in terms of the common cyclogenesis regions affecting eastern Asia. Yoshiike and Kawamura (2009) found that while bomb cyclogenesis occurred slightly more frequently during weak EAWM years, it was more concentrated along the south-eastern Japan coast during strong EAWM, owing to larger heat fluxes over the

Kuroshio current.

Locally-intense mesoscale cyclones have also been documented across the Sea of Japan, developing in response to polar outbreaks over the warmer waters in conjunction with the complex terrain along and north of the Korean peninsula. Tsuboki and Asai (2004) describe the process of strong convergence forming east of the Korean peninsula with substantial sensible and latent heating from the Sea of Japan leading to the

formation of these mesoscale cyclones. Intense Sea of Japan cyclones can cause substantial wave activity and subsequent coastal damage along the east coast of Korea (Lee and Yamashita, 2011; Oh and Jeong, 2014; Mitnik et al., 2011), in addition to significant snowfalls across Korea. Clearly, an accurate representation of air-sea interactions in NWP models is important when forecasting the impacts of winter cyclones and accompanying heavy snowfalls across the Korean peninsula.

It is indicated by numerous studies that data assimilation could help to obtain more accurate winter weather forecasts. Studies showed that in situ and remote sensed observations for surface conditions and the upper-atmosphere can provide a better description for both storm-scale processes and large-scale environments leading to improved precipitation forecasts (Zupanski et al., 2002; Cucurull et al., 2004; Zhang et al., 2006; Fillion et al., 2010; Hartung et al., 2011; Hamill et al., 2013; Salslo and Greybush, 2017; English

et al., 2018; Zhang et al., 2019). In South Korea, data assimilation also indicated significant benefit for winter forecast (Kim et al., 2013; Kim and Kim, 2017; Yang and Kim, 2021). For example, Kim et al. (2013) demonstrated the assimilation of the conventional surface and upper air observations, aircraft, and multiple



satellite observations located upwind or in the vicinity of the Korean peninsula into the Korea Meteorological Administration (KMA) Unified Model. The result showed large decreases in the forecast error for the 24-, 36-, and 48-h forecasts of a strong winter storm event.

It is indicated that better representation of air-sea interaction from the ocean can provide benefit to the forecast of winter storms occurred in the downstream regions. For example, Peevey et al. (2018) showed a significant reduction in forecast error when observations over Pacific Ocean were assimilated for winter storms in western United States. Therefore, it is of great interest to assimilate the observations over oceans surrounding the Korean peninsula and examine their impacts on winter storms affecting the peninsula. Since regular observations over these oceans are limited to only a few buoys, satellite observations and retrieved products that can provide a broad spatial coverage with regular revisit of the data sparse regions may be of substantial benefit. In support of the International Collaborative Experiments for PyeongChang 2018 Olympic and Paralympic Winter Games (ICE-POP 2018) field campaign, ocean surface meteorology conditions were retrieved from the Global Precipitation Measurement (GPM) microwave observations from January to March 2018. The motivations of the current research are to explore the utilization of this special dataset and examine the impact of the dataset on forecast of heavy snowstorms over the Korean peninsula using data assimilation.

## 2. Data and Methods

### 2.1 The GPM Retrieved ocean surface Meteorology Data for ICE-POP 2018

The ICE-POP 2018 field campaign was led by the KMA as a component of the World Meteorological Organization (WMO)'s World Weather Research Programme (WWRP) Research and Development and Forecast Demonstration Projects (RDP/FDP) in order to enhance the capability of convective scale numerical weather prediction modeling and to improve the understanding of the high impact weather systems. The field campaign took place during the Winter Olympics (February-March) of 2018 in support of the 23rd Olympic Winter held in PyeongChang, Korea on 9-25 February and the 13th Paralympic Winter Games in 9-18 March 2018 which ran in real-time to provide guidance to forecasters during the Olympic Games. The focuses of the ICE-POP 2018 were to collect observations to measure the physics of heavy snow over the complex terrain in the PyeongChang region of South Korea and to improve the predictability of winter storm forecasting. During the ICE-POP 2018, remote sensing and in situ observations were collected with an intensive instrument network including enhanced surface weather stations, radiosondes and wind profilers. Cloud and precipitation processes were observed with four KMA radars and an X-band radar, National Aeronautics and Space Administration (NASA) Dual Frequency Dual Polarimetric Doppler Radar (D3R), lidar, Precipitation Imaging Packages (PIP), Micro Rain Radars (MMR), Microwave Radiometers, Parsivel





disdrometers, etc. An aircraft and a marine weather observing ship also deployed during the campaign (as detailed in Petersen et al. (2018)).

As part of NASA Weather Focus Area and GPM support of the ICE-POP 2018 program, near-real-time ocean surface turbulence flux retrievals were produced based on Roberts et al. (2010) using intercalibrated

passive microwave radiometer observations that were produced in support the Integrated Multi-SatellitE Retrievals for GPM precipitation product (Berg et al., 2018). While intended to support precipitation estimation, these brightness temperatures are also capable of supporting the estimation of the marine surface meteorology — wind speed, sea surface temperature, air humidity and temperature — that are required to estimate the surface turbulent fluxes. The microwave imagers provide information on near-surface winds,

moisture, and temperature associated with the 10, 18.7, 23.8, 36.5, and 89[1] GHz vertical and horizonal polarized microwave channels. These channels are used together with an a priori estimate of sea surface temperature from the NCEP real-time global high-resolution (1/12º) sea surface temperature (RTG-SST) product to retrieve 10-m wind speed, 2-m specific humidity, and 2-m air temperature, and sea surface temperatures. The retrieval algorithm is based on a single-layer neural network following Roberts et al.

(2010). A large training dataset of standardized ocean buoy observations collocated within 1 hour and 25 km of observations with each microwave sensor was developed. These data were broken into a training and and set-aside independent validation dataset with a 60% and 40% split, respectively. For training data, the data was split into a training and cross-validation dataset with a 70% and 30% split. These retrieved parameters were then used to estimate the surface turbulent fluxes through application of the Coupled Ocean–

Atmosphere Response Experiment (COARE) 3.5 (Edson et al., 2013) bulk flux algorithm. In this paper, we are interested in assimilating the near-surface estimates directly rather than use of the fluxes. Compared to the independent validation data, the root-mean-square (RMS) uncertainties are assessed at 1.1 g kg$^{-1}$, 0.9 K, and 1.2 m s$^{-1}$ for surface humidity, temperature, and wind speed, respectively based on the mean statistics computed for GPM Microwave Imager (GMI), Advanced Microwave Scanning Radiometer 2 (AMSR2), and

the Special Sensor Microwave Imager/Sounder (SSMIS) microwave imagers for which retrievals were developed. The retrievals were essentially unbiased against the validation observations. The forcing model makes use of a high-resolution sea surface temperature estimate (the NASA Short-term Prediction Research and Transition (SPoRT) SST product) and those retrieved were not used.

The GPM-retrieved surface observations are generally available over the oceans around the Korean

peninsula within 1 h from 00, 06, 09, 12, 18, and 21 UTC on 7-8 March 2018. For February 27-28, the

---

[1] Not all microwave imagers share the same central frequencies. However, each has a comparable channel near each of the bands listed.





retrieved data are typically available within 1 h from 00, 06, 09, 15, 18, and 21 UTC. The coverage of the retrieval product varies with time due to the geolocation of the microwave imager swaths. At most of the above-mentioned times, observations typically cover ~27° - 50° N over the Sea of Japan and the western North Pacific Ocean to the east of Japan. At 09, 18 and 21 UTC, Bohai Sea and Yellow Sea to the west of

the Korean peninsula are usually observed or partly observed. Figure 1 shows an example of GPM-retrieved 2-m temperature, 2-m specific humidity, and 10-m wind speed at 09 UTC 7 March 2018 when the observations cover the west part of the Bohai Sea and Yellow Sea, most part of the Sea of Japan, and western North Pacific Ocean. As this time, cold (< -1 °C) and dry (< 3.0 g kg$^{-1}$) air was observed at latitude above 44° N and warm (> 17 °C) and moist (> 10.5 g kg$^{-1}$) air at latitude lower than 30° N. Observed surface

temperature ranges from -7 – 23 °C and surface humidity from 0 – 13.5 g kg$^{-1}$ over the model domain. Surface wind speed is found between 0 – 18 m s$^{-1}$. Low wind centers appeared near the northern coast of Japan, one in central east Sea of Japan and the other one in western North Pacific Ocean.

*2.2 Data Assimilation System and Numerical Experiments*

Two heavy snowstorms affecting the Korean peninsula and ICE-POP field domain on 27-28 February and 7-8 March 2018 were selected for case studies. Figures 2 and 3 summarize the evolution of surface features for the two case studies. In both instances, a surface low pressure developed to the south and southwest of the Korean peninsula and tracked to the northeast, passing along or just off the South Korea southern coast, placing the mountainous portions of South Korea, including the Olympics/Paralympics venue, in the

favorable northwestern quadrant of the surface low for heavy snowfall. In the 27-28 February snowstorm, a closed 1005-hPa low is situated just off the eastern China coastline at 0000 UTC 28 February, to the southwest of the Korean peninsula, with another closed low over northeastern China at 1008 hPa (Fig. 2a). The southern low experienced substantial deepening as it tracked northeastward over the next 24 to 36 hours, reaching extreme southern South Korea by 1200 UTC 28 February at 994 hPa intensity (Fig. 2b), the central

Sea of Japan by 0000 UTC 1 March at 987 hPa and absorbing the northern low by this time (Fig. 2c), and then into northern Japan by 1200 UTC 1 March at 974 hPa minimum central pressure (Fig. 2d). The 28 February was the warmer of the two snowstorms, with most snow accumulation confined to the mountainous terrain along the Korean east coast, including the Olympics venue, where storm-total snow accumulations of ~40 cm were observed (not shown). Gehring et al. (2020) analyzed the warm conveyer belt and microphysical

characteristics of this heavy precipitation event, using datasets from the ICE-POP 2018 field campaign.

Temperatures were slightly colder during the 7-8 March event, resulting in a more widespread snowfall across the southern and eastern Korean peninsula within the mountains and at lower elevations. Following the general synoptic snows, a surge of stronger north/northeasterly low-level winds off the Sea of Japan





affected the Korean east coast and eastern mountains, leading to enhanced residual precipitation and strong

orographic uplift (not shown). The 7-8 March extratropical cyclone began as a weak, open wave at 1200

UTC 7 March (Fig. 3a), then deepened to a 1010-hPa closed low to the southeast of the Korean peninsula at

0000 UTC 8 March (Fig. 3b). The cyclone slowly strengthened over the next 24 hours to 1004 hPa over the

eastern Sea of Japan by 1200 UTC 8 March (Fig. 3c) and then to 1003 hPa as it tracked northeastward into

northern Japan by 0000 UTC 9 March (Fig. 3d). An elongated meridional trough extended out of the low

pressure center across much of Japan, resulting in a long fetch of north/northeasterly low-level winds across

the Sea of Japan that affected the east coast of Korea during 8 March.

The Advanced Research Weather Research and Forecasting (WRF ARW; Powers et al., 2017) model was

used to conduct the regional simulations for the two events. The snowstorms were simulated using 3-nested

domains with horizontal resolution of 9-, 3-, and 1-km and 62 vertical levels as illustrated in Fig. 4. The

model physics options include the Goddard long-wave and shortwave radiation schemes (Chou and Suarez,

1999), Grell-Freitas cumulus parameterization (Grell and Freitas, 2014), Mellor-Yamada-Janjic (MYJ) PBL

schemes (Janjic, 1994), Morrison 2-moment microphysical scheme (Morrison et al., 2009) and Unified Noah

land-surface model (Chen and Dudhia, 2001). The cumulus parameterization was only used for the outer 9-

km resolution domain.

In the present study, the community Gridpoint Statistical Interpolation (GSI; Wu et al., 2002) v3.6 system

was used to assimilate the GPM-retrieved ocean surface meteorology data. The GSI system was initially

developed by the NCEP Environmental Modeling Center (EMC) and is currently maintained and supported

by the National Oceanic and Atmospheric Administration (NOAA) Development Testbed Center (DTC; Hu

et al., 2016). The GSI is built in physical space for a unified, flexible, and efficient modular system for

multiple parallel computing environments and has been implemented real-time into both global and regional

data assimilation (Wu, 2005; De Pondeca et al., 2007; Kleist, 2009). The community GSI is functionally

equivalent to the operational version used in NCEP. The system readily incorporates multiple types of

observational data including conventional data, radar, and satellite radiance and retrieved products.

The GSI system is a 3-dimensional variational (3DVAR) data assimilation system (more detailed

description in Wu et al., (2002)). In GSI, the 3DVAR cost function **J** is defined to measure the difference

between the model and observations as:

$$\mathbf{J} = \mathbf{J}_b + \mathbf{J}_o + \mathbf{J}_c = 1/2\ [\mathbf{x}^T\mathbf{B}^{-1}\mathbf{x} + (\mathbf{Hx} - \mathbf{y})^T\mathbf{R}^{-1}(\mathbf{Hx} - \mathbf{y})] + \mathbf{J}_c \qquad (1)$$

where **x** is the analysis increment ($\mathbf{x}_a - \mathbf{x}_b$), $\mathbf{x}_a$ is analysis fields, $\mathbf{x}_b$ is background fields, $\mathbf{J}_c$ is constraint terms,

**B** is the background error covariance matrix for analysis control variables, **y** is the observational residuals,

($\mathbf{y} = \mathbf{y}_{obs} - \mathbf{Hx}_{guess}$), **R** is the observational error covariance matrix, and **H** represents a transformation



operator from the control variables to the observations. The control variables in GSI include stream function, unbalanced velocity potential, unbalanced virtual temperature, unbalanced surface pressure, and pseudo relative humidity.

Table 1 lists the numerical experiments and corresponding data assimilation activities preformed for the two cases. Two different numerical experiments were conducted for each snowstorm event. For the March 7-8 case, the control experiment (CTRL_Mar) assimilates the conventional observations every 6-h using the prepbufr data obtained from the National Center for Atmospheric Research (NCAR) Research Data Archive (available at http://rda.ucar.edu/data/ds337.0). The conventional data refers to the global surface and upper air observation operationally collected by the National Centers for Environmental Prediction (NCEP) which includes surface, marine surface, radiosonde, pibal and aircraft reports from the Global Telecommunications System (GTS), profiler, United States radar derived winds, SSM/I oceanic winds and total precipitable water retrievals, and satellite wind data from the National Environmental Satellite Data and Information Service (NESDIS). Another experiment, DA_Mar, assimilates the GPM-retrieved ocean surface temperature, specific humidity, and wind speed observations besides the conventional data. As shown in Table 1, cycled assimilation of the GPM-retrieved ocean surface meteorology data was performed at 06, 09, 12, 18, and 21 UTC of 7 March and 00, 06, 09, 12, 18, and 21 UTC 8 March based on the availability of the retrieval product. Both experiments began at 00 UTC 7 March 2018 and ended at 00 UTC 9 March 2018. For both experiments, the initial and boundary conditions of the WRF background field were interpolated from the 0.5° resolution Global Forecast System (GFS) analysis. For the 27-28 February 2018 event, CTRL_Feb and DA_Feb were conducted with settings similar to CTRL_Mar and DA_Mar, respectively. Both experiments started at 00 UTC 27 February 2018 and ended at 00 UTC 1 March 2018. For DA_Feb, the GPM-retrieved ocean surface meteorology data was assimilated at 06, 09, 15, 18, and 21 UTC of 27 February and 00, 06, 09, 15, 18, and 21 UTC 28 February 2018.

## 3. Results

In this section, the data assimilation result is compared with WRF simulations and the observations collected for the March 7-8 and February 27-28 snowstorm cases. The impact of the data on initial conditions and short-term forecasts are examined.

*3.1. Case study for March 7-8 Snowstorm Event*

The scatterplot of 2-m temperature, 2-m specific humidity, and 10-m wind speed observations with respect to the departures between the observed values and WRF background (i.e., a positive departure represents a higher value in observation than the model) is shown in Fig. 5a-c. As a pre-process step before data assimilation, outliers with magnitude of surface temperature departure > 6 °C, specific humidity departure >


4 g kg$^{-1}$, or wind speed departure > 9 m s$^{-1}$ were removed. For the observations used for the two cases, the

median values are 10.59 °C, 6.47 g kg$^{-1}$, and 8.43 m s$^{-1}$ for surface temperature, specific humidity, and wind

speed. As shown in Fig. 5, the 25$^{th}$ (and 75$^{th}$) percentile values of the observed surface temperature, specific

humidity and wind speed are 4.96 (and 16.71) °C, 4.06 (and 8.70 g kg$^{-1}$), and 6.04 (and 10.74 m s$^{-1}$),

respectively. The probability density functions (PDF) of the departures of 2-m temperature, 2-m specific

humidity, and 10-m wind speed are shown in Fig. 5d-f. An apparent skewness to the positive side is shown

in the PDF of the surface temperature departure (Fig. 5d) with the mean (and standard deviation) of 1.19 (and

1.75) °C. For surface specific humidity, the departure shows a narrower spread with 95% of the values

ranging from -2 to 2 g kg$^{-1}$. The PDF of the surface specific humidity departure skews to the negative side

with the mean and standard deviation of -0.41 and 0.89 g kg$^{-1}$. This indicates a generally colder model

atmosphere with lower specific humidity at ocean surface in the WRF background when compared to the

observations. For surface wind speed, the departure has 98% of the values between -5 and 5 m s$^{-1}$. The mean

and standard deviation for the surface wind speed departure is 0.33 and 2.02 m s$^{-1}$, respectively.

Through data assimilation, the GPM-retrieved surface observation directly influences the thermodynamic

and wind fields of the WRF initial condition. Figure 6 displays surface condition of the March 7-8 case before

and after the data assimilation cycle at 09 UTC 7 March 2018. The A – B (difference between the data

assimilation analysis and model background, Fig. 6c, 6f, and 6i) was compared with O – B (difference

between the observation and model background, Fig. 6b, 6e, and 6h) to indicate the changes in surface

temperature, specific humidity, and wind speed fields by data assimilation. Before the data was assimilated,

surface temperature in background was generally lower than observation over Bohai Sea, Yellow Sea, and

Sea of Japan which is reflected by the areas of positive O – B with the magnitude up to 6 °C (Fig. 6b). High

O – B values were also found over western North Pacific Ocean at latitudes above 38° N. After data

assimilation, an increase was made in surface temperature indicated by positive A – B over Bohai Sea and

Yellow Sea, generally positive A – B over Sea of Japan and western North Pacific Ocean (Fig. 6c) where

positive O – B was found. For surface specific humidity, the background was mostly lower than the

observation over northern and central Sea of Japan. Over western North Pacific Ocean, O – B was generally

positive at latitudes above 38° N. A large area of negative O – B with magnitude down to -3 g kg$^{-1}$ were

found at latitudes lower than 38° N (Fig. 6e). After data assimilation, surface moisture has been enhanced

with overall positive A – B over Bohai Sea, Yellow Sea, Sea of Japan as well as in western North Pacific

Ocean at latitude above 36° N (Fig. 6f). For surface wind speed, the background is apparently lower than

observation at northern Sea of Japan and northern part of western North Pacific Ocean and generally higher

than observation at central to southern Sea of Japan and southern part of western North Pacific Ocean for

latitude below 40° N (Fig. 6h). After data assimilation, A – B generally agrees with the pattern shown in O –



B with wind speed increase at regions with positive O – B and decrease at regions with negative O – B (Fig. 6i).

The impact of the GPM-retrieved surface meteorology data accumulates with the data assimilation cycles.

Figure 7a shows surface temperature difference between DA_Mar and CTRL_Mar after the 5th data assimilation cycle and Fig. 7e shows O – B of surface temperature before data assimilation. The positive O – B (Fig. 7e) over Bohai Sea and Yellow Sea, northern Sea of Japan, and western North Pacific Ocean explains the positive surface temperature increment at the corresponding and surrounding regions shown in Fig. 7a. Different from Fig. 6c in which the data impact was mainly concentrated over ocean at Bohai Sea,

Yellow Sea, Sea of Japan and western North Pacific, the impact of the GPM-retrieved data in Fig. 7a has spread over the entire model domain with 5 cycles of data assimilation. Numerical integration during 06 to 21 UTC with modified initial conditions is a large contributor to this difference especially for the model grids over land that are far from the observed locations (e.g., the area of large positive departure created in the northwest corner of the model domain around 112-120º E). Fig. 7b-d displays 1-, 2-, and 3- h forecasts using

the data assimilation analysis at 21 UTC 7 March 2018 as the initial condition of WRF model. An important indication from Fig. 7b-d is the strong adjustment in the model field with time integration. For example, the data assimilation analysis field shows several large areas of positive departure above 2 ºC in Yellow Sea and western North Pacific Ocean (Fig. 7a) with a maximum value of 4.5 ºC in northern Yellow Sea. Within the first hour of integration, the magnitudes of the positive departure over these regions have been largely reduced

with the maximum value at Yellow Sea dropped to 3.2 ºC (Fig. 7b). The size of the area over these regions with departure above 2 ºC have also greatly shrank. Further adjustment was found in 2-h and 3-h forecast fields (Fig. 7c,d), but with a much smaller magnitude of change when compared to the change in the first hour of integration. In addition, the large positive departure over the northwest corner of the model domain near 112-120º E became higher with 3 hours of integration (Fig. 7d), another indicator of the model

adjustment.

The strong model adjustment shown in Fig. 7 was also seen in moisture and wind speed fields. Fig. 8a shows the root-mean-square difference (RMSD) in surface temperature, surface specific humidity, and surface wind speed between DA_Mar and CTRL_Mar calculated over the entire model domain for 0-6 h forecast after the 5th data assimilation cycle conducted at 21 UTC 7 March 2018. At 21 UTC, domain-

averaged RMSD is 1.14 ºC, 0.41 g kg$^{-1}$, and 1.72 m s$^{-1}$ for surface temperature, specific humidity, and wind speed, respectively. After the first hour of integration, the RMSD showed a rapid decline to 0.95 ºC, 0.35 g kg$^{-1}$, and 1.66 m s$^{-1}$ which is 17%, 15%, and 4% reduction of the original values at the analysis time. In the next 5 hours' integration, these values dropped slowly to 0.78 °C, 0.29 g kg$^{-1}$, and 1.59 m s$^{-1}$, corresponding to additional 14%, 15%, and 4% reduction. Fig. 8b-d provides the vertical profiles of RMSD of temperature,



specific humidity, and wind speed calculated over the entire model domain. Profiles T, Q, and WSPD
represent the RMSD values at 61 model vertical levels at 21 UTC and T1, Q1, WSPD1 are for 22 UTC 7
March 2018. After 1 h time integration, Fig. 8b shows a sharp decrease in temperature RMSD at low level
atmosphere below model level 20 (~ 850 hPa). For specific humidity, Fig. 8c indicates a decrease in RMSD
in boundary layer below model level 8 (~ 925 hPa) and an increase in mid-level atmosphere from model level

20 to 38 (~ 600 hPa). From 21 UTC to 22 UTC, a consistent increase in the wind speed RMSD was found
(Fig. 8d) from model level 13 (~ 900 hPa) to high level atmosphere at model level 50 (~ 300 hPa).

The impact of data assimilation has been examined by comparing the result from the numerical
experiments with the South Korean Surface Analysis dataset based on Ryu et al. (2020). The South Korean
Surface Analysis is a product interpolated from the observations collected by Automatic Weather Stations

(AWS) in South Korea using a radial basis function (Ryu et al., 2020). This dataset is in Lambert Conic
confoemal project with 1-km spatial resolution and 10-min time interval. Figure 9 shows 1-h precipitation
observed by the South Korean Surface Analysis at 16 UTC 7 March 2018 when compared with the results in
CTRL_Mar and DA_Mar. At this time, light snowfall was broadly observed over northern to central South
Korea. The storm started to produce heavier snowfall in the southern region with >3 mm h$^{-1}$ in the

southwestern tip of the Korean peninsula and >8 mm h$^{-1}$ in Jeju Island. From Fig. 9b and 9c, it is indicated
that the simulated storm in both DA_Mar and CTRL_Mar also produced light to moderate snowfall over
most area of South Korea with heavier precipitation above 3 mm h$^{-1}$ over the South Jeolla Province. The
pattern of precipitation in DA_Mar is very similar to CTRL_Mar. Strong precipitation above 10 mm h$^{-1}$ was
predicted in Jeju Island in both CTRL_Mar and DA_Mar. Comparing to DA_Mar, CTRL_Mar produced an

overall stronger precipitation indicated by the larger area with precipitation rate above 1 mm h$^{-1}$ from central
to southern South Korea.

This system moved northeastward and precipitation related to the system firstly appeared in the southwest
end of South Korea at 08 UTC 7 March. Large amount of precipitation in South Korea concentrated between
17 UTC 7 and 00 UTC 8 March. In Fig. 10, 24-h accumulated precipitation from 06 UTC 7 to 06 UTC 8

350   March is plotted for the observation and compared with the numerical experiments. From the South Korean
Surface Analysis, there were over 20 mm snowfall produced in the southern provinces and cities with over
40 mm snowfall along the coast in South Gyeongsang Province. As shown in Fig. 10b and 10c, both
CTRL_Mar and DA_Mar overpredicted the precipitation amount with >15 mm precipitation covered almost
the entire South Korea. Overestimation is especially apparent in CTRL_Mar which predicted precipitation

of >40 mm over most area of the central and southern provinces. Heavy snowfall above 70 mm with the
extreme value of 95 mm was predicted along the southeastern coast of Ulsan and North Gyeongsang province
in CTRL_Mar. In comparison, DA_Mar predicted a much smaller area with precipitation exceeding 40 mm.



In addition, the maximum precipitation was 82 mm over coast of Ulsan, 13 mm lower than CTRL_Mar. The Probability Density Function (PDF) of 24-h precipitation amount was calculated with bin-size of 2 mm over

South Korea where the surface analysis data is available. It is shown that 13.5% of the observed area in South Korea had light precipitation of 0-2 mm. 77.2% of the entire area observed snowfall of <20 mm, 19.6% of the area had precipitation of 20-40 mm, and only 3.2 % of the area experienced precipitation of >40 mm in the 24 hours. Both CTRL_Mar and DA_Mar predicted much lower percentage of area (i.e., 2.5% and 2.2%, respectively) for light precipitation of 0-2 mm. In CTRL_Mar, the peak precipitation probability was

predicted at 42 mm for 6.7% of the area in South Korea. Snowfall of <20 mm occurred at 19.3% of the entire area which is 57.9% smaller than the observation. 20-40 mm snowfall was predicted in CTRL_Mar for 39.4% of the area, 19.8% higher than the observed value. Precipitation of >40 mm was generated for 41.3% of the area which is 38.1% larger than the observation. DA_Mar performed better than CTRL_Mar with an overall lower precipitation amount. The peak precipitation probability was at 36 mm for 8.8% of the area. Light

snowfall below 20 mm was predicted for 23% of the area, 20-40 mm snowfall for 60% of the area, and >40 mm for 17% of the area, indicating less overprediction. Besides the model validation, it is also of our interest to understand the accuracy of GPM observed precipitation for snowstorms over the Korean peninsula. Figure 10d shows 24-h accumulated precipitation between 06 UTC 7 and 06 UTC 8 March from Level 3 Integrated Multi-satellitE Retrievals for GPM (IMERG) Final Run. Comparing with the South Korean Surface Analysis

data, IMERG produced comparable amount of precipitation in northern South Korea and overall larger precipitation amount over central South Korea. The strong precipitation center of >35 mm in IMERG was to the northeast of the one observed in South Gyeongsang Province by South Korean Surface Analysis. An overestimated precipitation center exceeding 35 mm was also presented in west coast of South Korea by IMERG. These features were reflected in the PDF of IMERG precipitation by the much larger probability

for precipitation from 20 to 40 mm and much smaller probability for precipitation from 2 to 14 mm.

Positive impact of the GPM-retrieved data was also found in surface temperature forecast with cycled data assimilation. The Root-Mean-Square error (RMSE) of 2-m temperature in Table 2 was calculated every 6-h across South Korea from 12 UTC 7 March to 00 UTC 9 March using the South Korean Surface Analysis as the reference dataset. From 12 UTC 7 to 00 UTC 8 March, the RMSE values in DA_Mar were close to

those in CTRL_Mar (i.e., 0-0.01 °C difference). After the 7[th] cycle of data assimilation at 06 UTC 8 March, RMSEs in DA_Mar were consistently smaller than CTRL_Mar. At the end of the model simulation time, surface temperature RMSE in DA_Mar was 2.14 which is 0.33 °C lower than CTRL_Mar. Figure 11 shows an example of 2-m temperature from the South Korean Surface Analysis compared with CTRL_Mar and DA_Mar at 15 UTC 8 March 2018. Generally, surface temperature was around 0-4 °C at this time over South

Korea with a few warmer areas of 4-6 °C along the southeastern coast and Jeju Island. Colder temperature (-





2°C) was observed in the Taebaek Mountains with a few spots below -4 °C (Fig. 11a) in the northern tip of Gangwon province. Both CTRL_Mar and DA_Mar predicted colder temperature than the observation in South Korea. This is especially apparent along the Taebaek Mountain range where temperature below -4 °C was produced in DA_Mar and below -6 °C in CTRL_Mar. Over Sobaek Mountains, temperature below -2

°C was produced in DA_Mar and below -4 °C in CTRL_Mar, which were 2-4°C colder than the observation. Comparing with DA_Mar, CTRL_Mar produced a much larger area with temperature below 0 °C and apparently colder temperature along the Taebaek Mountains.

*3.2. Case Study for February 27-28 Snowstorm Event*

The above results indicate a positive impact of the GPM-retrieved surface meteorology data on the short-term forecast of the March 7-8 case. The assessment of data assimilation result for the February case will be discussed in this section.

Figure 12 shows the surface condition in model background compared with O – B and A – B at 09 UTC 27 February 2018. The GPM-retrieved observation at this time is available for a large part of Bohai Sea and

the eastern half of Yellow Sea and East China Sea. The observation also covers Sea of Japan and western North Pacific Ocean. From Figs. 12b and 12e, the observed surface air was generally warmer and more humid in Bohai Sea and colder and drier in Yellow Sea than the model background. Over most of the regions in Sea of Japan and western North Pacific Ocean, surface air was roughly >1 °C warmer but with lower specific humidity than the model background. After data assimilation, positive increments in surface temperature and

specific humidity were produced over Bohai Sea, Sea of Japan, as well as western North Pacific Ocean with maximum increase of 3.5 °C and 1.2 g kg$^{-1}$. In Yellow Sea, negative increments in surface temperature and specific humidity were produced when the data was assimilated (Fig. 12c,f). For surface wind speed, GPM-retrieved observation (Fig. 12h) was generally lower than the background over Bohai Sea, Yellow Sea, and East China Sea, and higher in Sea of Japan and western North Pacific Ocean. After data assimilation, areas

of decreased wind speed were found in Bohai Sea, Yellow Sea, and East China Sea. Areas of increased surface wind speed have been produced in western North Pacific Ocean with magnitude up to 3.5 m s$^{-1}$.

Figure 13 displays 24-h accumulated precipitation from 21 UTC 27 to 21 UTC 28 February 2018 observed by South Korean Surface Analysis and compared with the results from CTRL_Feb and DA_Feb. It is indicated in Fig. 13a that widespread precipitation >15 mm was generated across South Korea in 24-h with

extreme precipitation of 184 mm over Jeju Island. Snowfall over 40 mm was produced in the 24 hours along the southeast coast of South Korea and the northeast coast above 36.5° N. From Fig. 13b and 13c, the precipitation pattern in DA_Feb is similar to CTRL_Feb. Both CTRL_Feb and DA_Feb produced widespread precipitation with regions of snowfall over 40 mm in central South Korea and along the east coast and the





south coast. When compared with the South Korean Surface Analysis, both CTRL_Feb and DA_Feb

overestimated the precipitation amount which is reflected by the much larger size of the areas with

precipitation over 40 mm. Comparing to CTRL_Feb, the area of snowfall over 40 mm in DA_Feb was smaller

and the extreme values in heavy precipitating centers were also lower. This overestimation is illustrated in

the PDF of precipitation in Fig. 13d. The highest precipitation probability was at 22 mm for 6.6% of the area

in observation, 28 mm for 7.8% of the area in CTRL_Feb, and 26 mm for 8.6% of the area in DA_Feb. It

was observed that 55% of the entire area with 0-20 mm precipitation, 40% of the area with 20-40 mm, and

5% of the area with > 40 mm. Both CTRL_Feb and DA_Feb predicted less area of light snowfall and more

area of moderate to heavy snowfall. This is especially apparent in CTRL_Feb. Light snowfall of 0-20 mm

was predicted for 26% of the area which is 29% lower than observation. 59% of the area was predicted to

have 20-40 mm snowfall, indicating 19% higher than observation. There were 15% of the area with > 40 mm

precipitation which is 10% higher than the observed value. DA_Feb generally performed better than

CTRL_Feb with higher occurrence of light to moderate snowfall and lower occurrence of heavy snowfall.

30% of the area was predicted with 0-20 mm, 61% of the area with 20-40 mm, and 9% of the area with

snowfall of >40 mm. Figure 13d shows GPM IMERG Final 24-h accumulated precipitation from 21 UTC 27

to 21 UTC 28 February. IMERG precipitation was above 35-40 mm over most area of South Korea, indicating

an apparent (~10 mm) widespread overestimate of precipitation for this snowstorm when comparing with the

South Korean Surface Analysis. In the plot of PDFs of precipitation, this is reflected by the much larger

probability for precipitation from 35 to 80 mm and much smaller probability for precipitation from 0 to 28

mm.

Even though the positive impact of data assimilation is not as significant as the one shown in the March

case, it is found that surface temperature forecast was improved for the February 27-28 event with data

assimilation. The RMSEs of 2-m temperature calculated over South Korea from 12 UTC 27 February to 00

UTC 01 March 2018 were listed in Table 2. At 12 UTC 27 February, the RMSE in DA_Mar equals that in

CTRL_Feb. From 18 UTC 27 to 00 UTC 28 February, RMSEs in DA_Feb were very close to (0.01 °C

difference) CTRL_Feb. From 06 UTC to 18 UTC 28 February, DA_Feb provided smaller RMSEs than

CTRL_Feb, implying more skillful forecast. At the end of the simulation period, both CTRL_Feb and

DA_Feb produced a larger RMSE than most of the previous times but with a smaller value in CTRL_Feb

than DA_Feb (2.59 vs. 2.64). Figure 14 shows 2-m temperature from the South Korean Surface Analysis

compared with forecasts from CTRL_Feb and DA_Feb at 18 UTC 28 February 2018. Generally, surface

temperature over northern to central South Korea was around 0-4 °C at this time and 4-6 °C in the southern

region. Warmer temperature of 6-8 °C was observed along Southern coast. Colder temperature (<-2 °C) was

observed along the Taebaek Mountains. Temperature forecast in CTRL_Feb was quite similar to DA_Feb at





this time. In the southern regions, predicted temperature was generally >2 °C colder than the observation. The apparent features also include the elongated area with cold air along the Taebaek Mountain range where temperature below -2 °C was produced in DA_Feb and below -4 °C in CTRL_Feb. DA_Feb outperformed

CTRL_Feb with a smaller area of cold temperature with smaller extreme value along the Taebaek Mountains.

**4. Discussion and Conclusion**

In this research, the GPM-retrieved ocean surface meteorology data has been assimilated with the community GSI v3.6 for two winter storm events during the ICE-POP 2018 field campaign. WRF ARW model

simulations were conducted for the two cases to investigate the impact of the retrieved data on winter storm forecast. The goal of this study is to assess and document the impact of this GPM-retrieval product on the forecast of winter systems over South Korean region.

The results indicate that large impact of the retrieved surface meteorology data has been produced on surface temperature, moisture, and wind speed fields in the initial condition (Figs. 6 and 12). While clear

biases remain in the model analyses over land, the assimilation of the surface meteorology does act to bring the analysis closer to the surface observations. These impacts are driven by the accumulation and spread of over-ocean surface meteorology information to the entire domain during cycled data assimilation. Strong model adjustment was found within 1-3 hours after data assimilation, which is commonly seen when surface data only is assimilated into the initial condition. Positive impact of the data on precipitation and temperature

forecasts was found for both cases when compared to South Korean Surface Analysis. Larger positive impact was found for the March 7-8 case (Figs. 9-11 and Table 2), while the impact of the retrieval product was slightly smaller for the February 27-28 case (Figs. 13-14 and Table 2). Although the synoptic pattern of both winter events belongs to warm low type, the synoptic configurations of the two surface low pressure systems were unlike. Different kinematic and thermodynamic processes involved in the two cases could be a major

contributing factor for the difference in data assimilation effect. Also, since observational data varies with time and only covers a part of Bohai Sea, Yellow Sea, Sea of Japan and western North Pacific Ocean at most of the times, the undersampling of retrieved product to the individual storm and important atmospheric structures critical to the development of the storm may be another important reason for the different effect of data assimilation.

The conclusions in this study are only based on two case studies. More experiments and continuous data assimilation with more cases are required to produce a more thorough evaluation of the statistical significance of the GPM-retrieved surface meteorology data assimilation. Furthermore, temperature departure was seen to be skewed (Fig. 5d) to the positive side and specific humidity departure to the negative side (Fig. 5e). Outliers were seen in the retrieved surface temperature, specific humidity, and wind speed data. It might be





appropriate to adopt a more rigorous gross check and data validation to retain valid data and remove observations that are largely skewed from the normal distribution of background departures. In future studies, more examinations and tests in the quality control and potentially adding a bias correction to the retrieved surface condition data may be a way to bring further improvement in precipitation forecast. In addition, the present study assimilated the GPM-retrieved surface meteorology data only with the conventional data. It is

of great interest to assimilate this observational type together with other types of observations obtained from the ICE-POP mission (e.g., radiosondes, radar observations, wind profilers) in which more information above the surface was included. This may provide a clearer understanding on how this type of observation could help to better depict the true state of atmosphere and hence benefit precipitation forecast.

**Code and data availability.** The community WRF model and the GSI data assimilation system can be accessed at https://www2.mmm.ucar.edu/wrf/users/ and https://dtcenter.org/com-GSI/users/, respectively. GPM satellite rainfall data is available at https://pmm.nasa.gov/data-access/downloads/gpm. The prepbufr data can be downloaded from http://rda.ucar.edu/data/ds337.0. The GPM-retrieved ocean surface meteorology data used for this research are publicly available at

https://www.nsstc.uah.edu/public/xuanli.li/GMD/. Any additional information related to this paper may be requested from the corresponding author.

**Author contributions.** Authors XL, WAP, JBR, JLC, CRH developed the research idea and modeling and data assimilation framework. XL and JS developed the data assimilation procedure and conducted the

experiments. JBR provided and processed the GPM-retrieved surface meteorology data. XL analyzed the results and WAP, JLC, CRH assisted in interpretation of the results. GL provided the South Korean Surface Analysis dataset and assisted JLC with analysis of the events. All authors participated in writing of the manuscript.

**Competing interests.** The authors declare that they have no conflict of interest.

**Acknowledgments.** We gratefully acknowledge funding provided for this research by the Earth Science Division (Dr. Tsengdar Lee) at NASA Headquarters as part of the NASA Short-term Prediction Research and Transition Center at Marshall Space Flight Center and the NASA Ground Validation component of the

NASA-JAXA GPM mission (Dr. Gayle Skofronick-Jackson). The authors would like to thank Drs. GyuWon Lee and Soorok Ryu at Kyungpook National University for providing the South Korean Surface Analysis dataset. The authors would like to express their appreciations to the participants in the World Weather





Research Programme Research Development Project and Forecast Demonstration Project, International Collaborative Experiments for Pyeongchang 2018 Olympic and Paralympic winter games (ICE-POP 2018)

hosted by the Korea Meteorological Administration.

**Financial support.** This research was supported by the Earth Science Division (Dr. Tsengdar Lee) at NASA Headquarters as part of the NASA Short-term Prediction Research and Transition Center at Marshall Space Flight Center and the NASA Ground Validation component of the NASA-JAXA GPM mission (Dr. Gayle

Skofronick-Jackson).





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



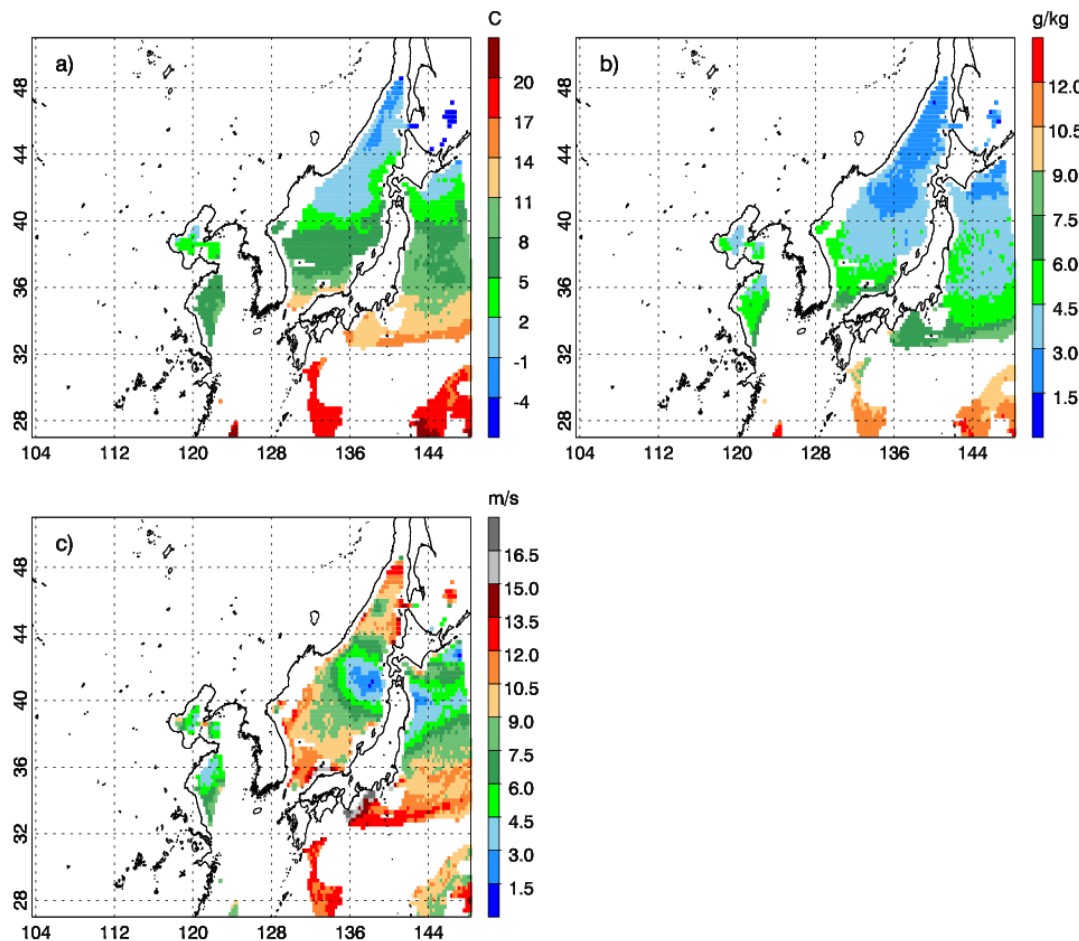


**Figure 1. A sample plot of the GPM-retrieved ocean surface meteorology data for a) 2-m temperature [°C], b) 2-m specific humidity [g kg⁻¹], and c) 10-m horizontal wind speed [m s⁻¹] at 09 UTC 7 March 2018.**







**Figure 2.** Eastern Asia regional surface meteorological analyses by the KMA at 12-hourly intervals, valid at (a) 00 UTC 28 February, (b) 12 UTC 28 February, (c) 00 UTC 1 March, and (d) 12 UTC 1 March 2018.




**Figure 3.** Same as Fig. 2, except valid at (a) 12 UTC 7 March, (b) 00 UTC 8 March, (c) 12 UTC 8 March, and (d) 00 UTC 9 March 2018.






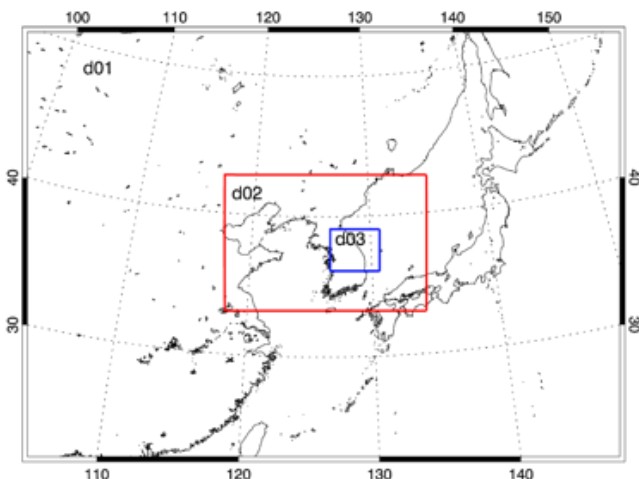

**Figure 4. WRF model domain configuration.**



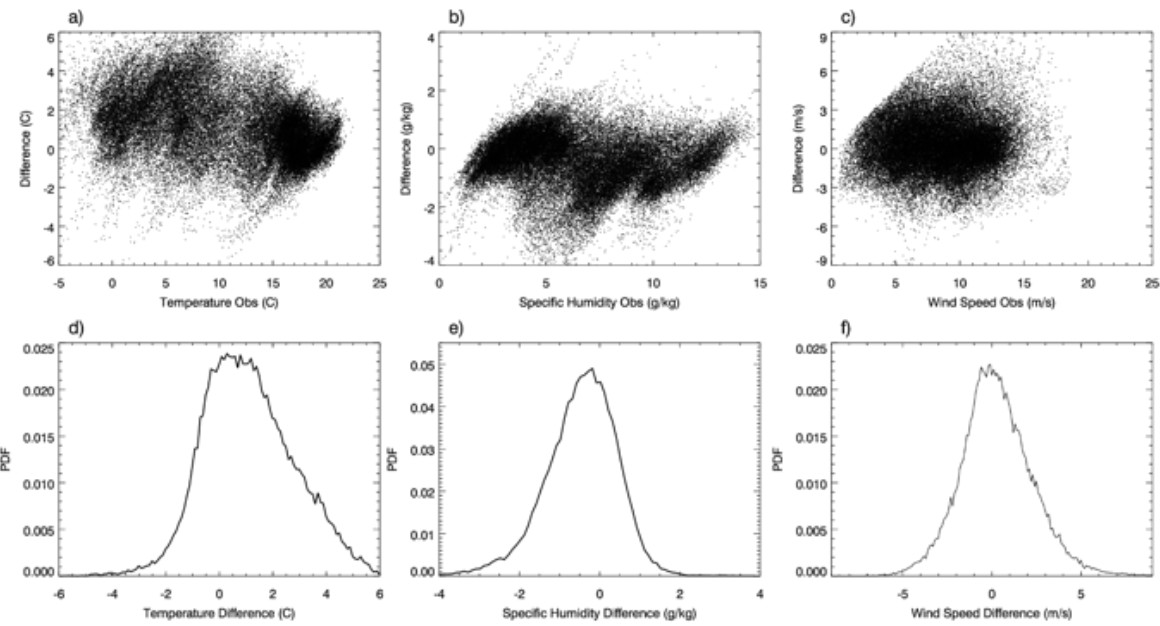

**Figure 5. Scatter plot for GPM-retrieved observation vs. the departure between the observation and the model background for a) 2-m temperature [°C], b) 2-m specific humidity [g kg⁻¹], and c) 10-m horizontal wind speed [m s⁻¹] from all available data during the data assimilation time windows for the two case studies. Probability density function (PDF) of the d) 2-m temperature departure, b) 2-m specific humidity departure, and c) 10-m horizontal wind speed departure with bin width of 0.1.**


**Figure 6. WRF Model background for a) 2-m temperature [°C], d) 2-m specific humidity [g kg$^{-1}$], and g) 10-m horizontal wind speed [m s$^{-1}$], difference between observation and model background for b) 2-m temperature, e) 2-m specific humidity, and h) 10-m horizontal wind speed, and difference between data assimilation analysis and the background for c) 2-m temperature, f) 2-m specific humidity, and**

**i) 10-m horizontal wind speed at 09 UTC 7 March 2018.**



**Figure 7. Difference between DA_Mar and CTRL_Mar in 2-m temperature [ºC] at a) 21 UTC, b) 22 UTC, c) 23 UTC 7 March 2018, and**

**d) 00 UTC 8 March 2018 and e) difference between observation and model background in 2-m temperature at 21 UTC 7 March 2018.**



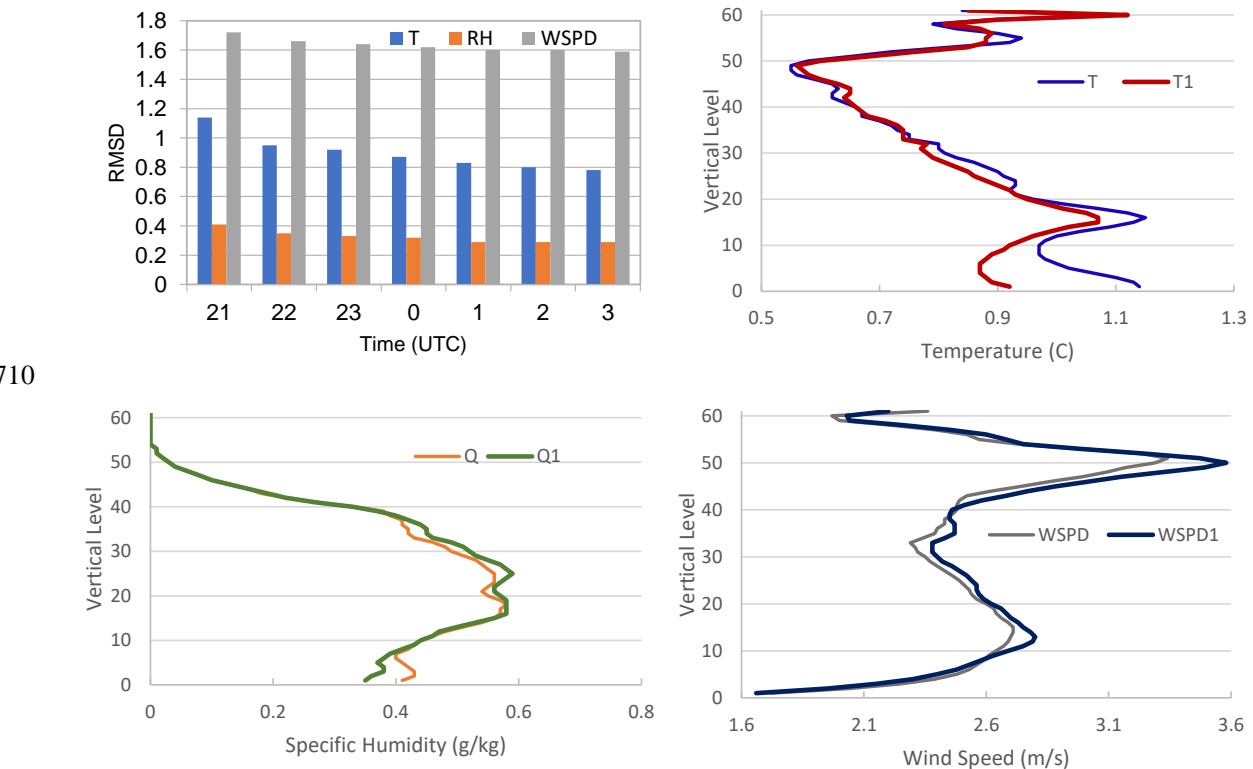


**Figure 8. a) RMSD between DA_Mar and CTRL_Mar for 2-m temperature [°C], 2-m specific humidity [g kg⁻¹], and 10-m horizontal wind speed [m s⁻¹] calculated over the model domain from 21 UTC 7 to 03 UTC 8 March 2018, and vertical profiles of RMSD for b) temperature, c) specific humidity, and d) wind speed at 21 UTC (T, Q, and WSPD) and 22 UTC (T1, Q1, and WSPD1) 7 March 2018.**






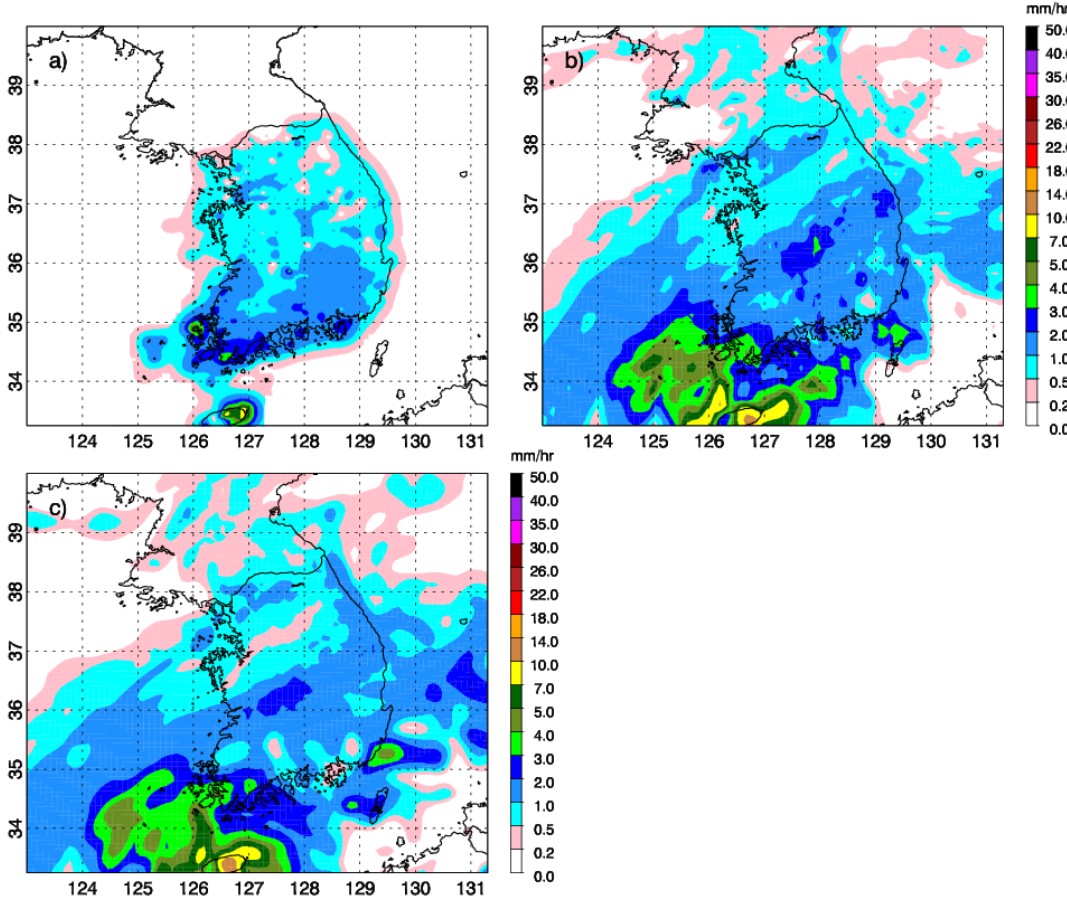

**Figure 9. 1-h precipitation (mm h$^{-1}$) from a) South Korean Surface Analysis, b) CTRL_Mar and c) DA_Mar at 16 UTC 7 March 2018.**






Figure 10. 24-h accumulated precipitation (mm) from a) South Korean Surface Analysis, b) CTRL_Mar, c) DA_Mar, and d) IMERG

Final from 06 UTC 7 March to 06 UTC 8 March 2018 and e) probability density function of 24-h accumulated precipitation.




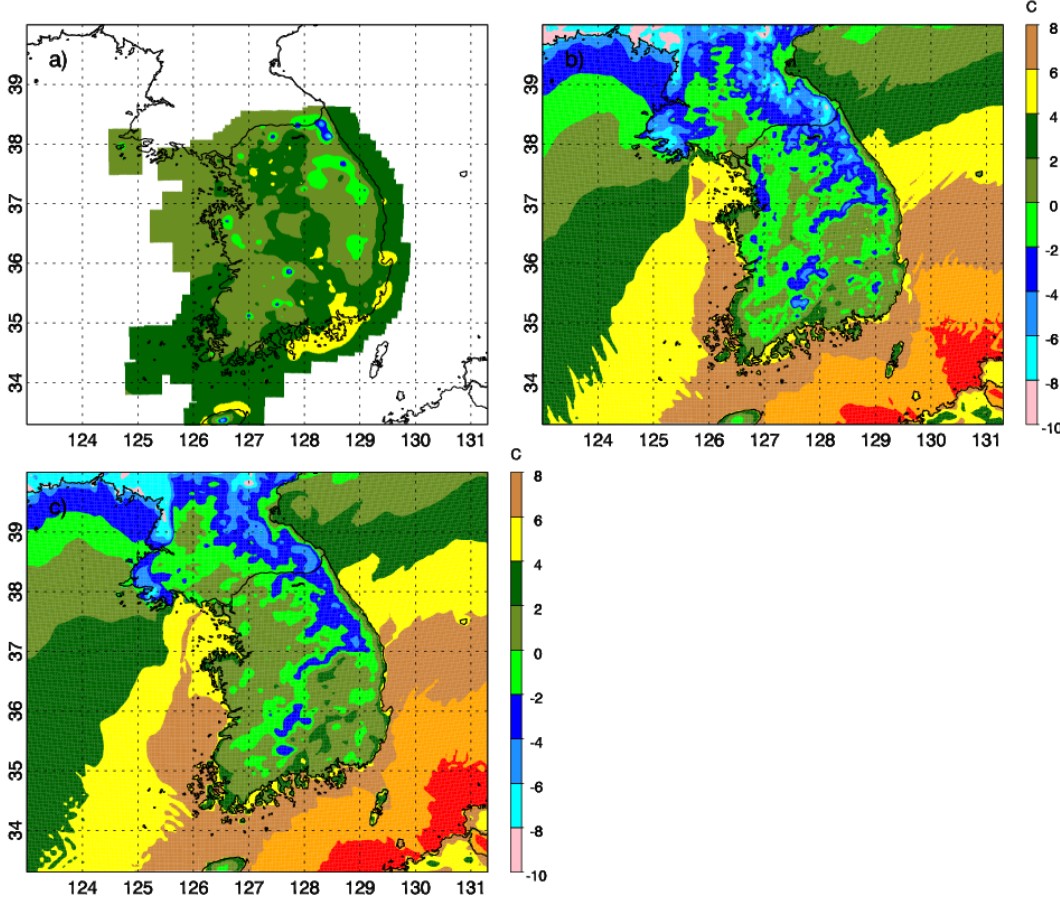

**Figure 11. 2-m temperature field [°C] from a) South Korean Surface Analysis, b) CTRL_Mar, and c) DA_Mar at 15 UTC 8 March 2018.**






**Figure 12. WRF Model background fields for a) 2-m temperature [°C], d) 2-m specific humidity [g kg⁻¹], and g) 10-m horizontal wind speed [m s⁻¹], difference between GPM-retrieved observation and model background fields for b) 2-m temperature, e) 2-m specific humidity, and h) 10-m horizontal wind speed, and difference between data assimilation analysis and model background filed for c) 2-m temperature, f) 2-m specific humidity, and i) 10-m horizontal wind speed at 09 UTC 27 February 2018.**



**Figure 13. 24-h accumulated precipitation (mm) from a) South Korean Surface Analysis, b) CTRL_Feb, c) DA_Feb, and d) IMERG**

**Final from 21 UTC 27 to 21 UTC 28 February 2018 and e) probability density function of 24-h accumulated precipitation.**




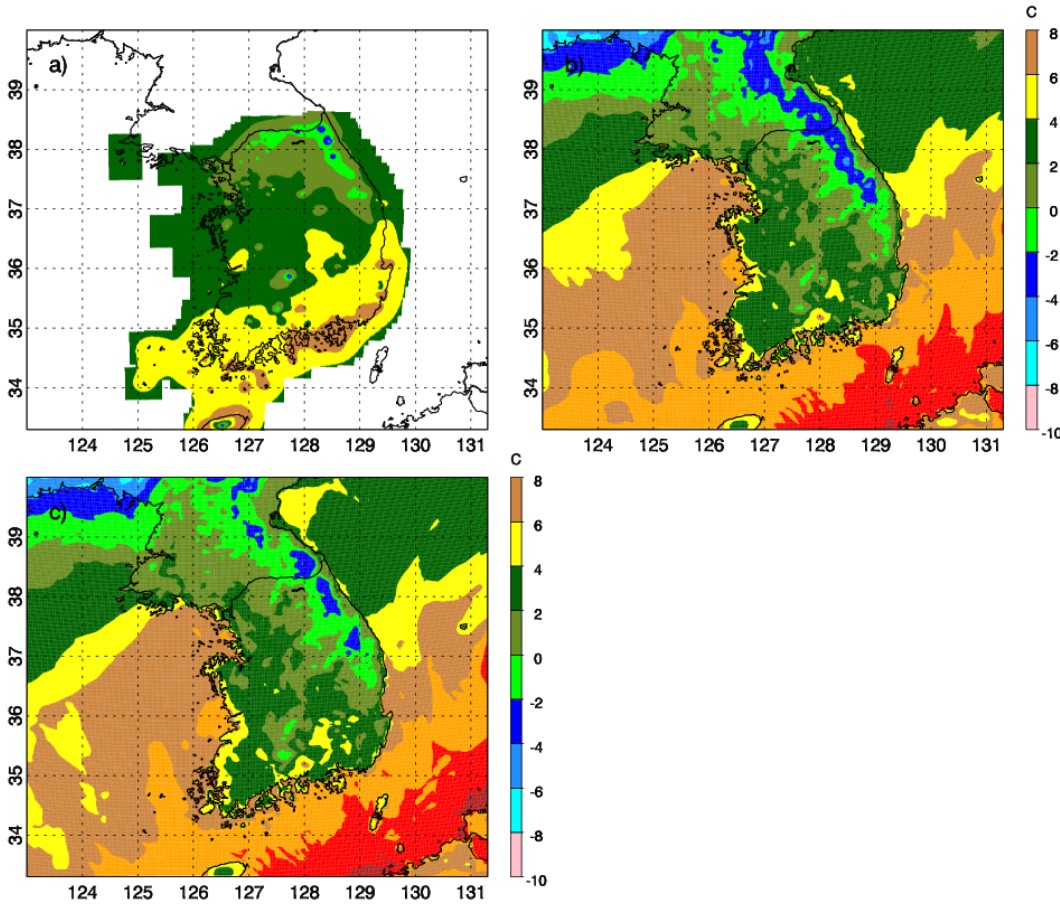

**Figure 14. 2-m temperature field [°C] from a) South Korean Surface Analysis, b) CTRL_Feb, and c) DA_Feb at 18 UTC 28 February 2018.**






**Table 1. Numerical experiments setup.**

| Experiment | Data | Data Assimilation Time |
|---|---|---|
| CTRL_Mar | Conventional data | 06, 12, and 18 UTC 7 March 2018 and 00, 06, 12 and 18 UTC 8 March 2018 |
| DA_Mar | GPM-retrieved surface meteorology data + conventional data | 06, 09, 12, 18, 21 UTC 7 March 2018 and 00, 06, 09, 12, 18, 21 UTC 8 March 2018 |
| CTRL_Feb | Conventional data | 06, 12, and 18 UTC 27 February 2018 and 00, 06, 12 and 18 UTC 28 February 2018 |
| DA_Feb | GPM-retrieved surface meteorology data + conventional data | 06, 09, 15, 18, 21 UTC 27 February 2018 and 00, 06, 09, 15, 18, 21 UTC 28 February 2018 |






**Table 2. Root-Mean-Square Error (RMSE) for 2-m temperature [°C] forecast over South Korea for difference experiments.**

| Time | CTRL_Mar | DA_Mar | Time | CTRL_Feb | DA_Feb |
|---|---|---|---|---|---|
| 12 UTC 03/07/2018 | 2.65 | 2.64 | 12 UTC 02/27/2018 | 1.95 | 1.95 |
| 18 UTC 03/07/2018 | 2.82 | 2.82 | 18 UTC 02/27/2018 | 2.10 | 2.11 |
| 00 UTC 03/08/2018 | 2.73 | 2.73 | 00 UTC 02/28/2018 | 2.33 | 2.32 |
| 06 UTC 03/08/2018 | 2.33 | 2.27 | 06 UTC 02/28/2018 | 2.11 | 2.06 |
| 12 UTC 03/08/2018 | 2.15 | 2.01 | 12 UTC 02/28/2018 | 2.70 | 2.52 |
| 18 UTC 03/08/2018 | 2.85 | 2.36 | 18 UTC 02/28/2018 | 2.29 | 2.06 |
| 00 UTC 03/09/2018 | 2.47 | 2.14 | 00 UTC 03/01/2018 | 2.59 | 2.64 |