# Peer review of "Assimilation of GPM-retrieved Ocean Surface Meteorology Data for Two Snowstorm Events during ICE-POP 2018"

_Geoscientific Model Development, 2021_

## Referee Comment (RC2)

**Review of manuscript**

**"Assimilation of GPM-retrieved Ocean surface Meteorology Data for Two Snowstorm Events during ICE-POP 2018"**

**by Li et al.**

**General assessment:**
The manuscript describes the assimilation of GPM-retrieved near-surface variables into the WRF model for two snow-storm case studies during the ICE-POP 2018 research and forecast programs during the Olympic and Paralympic Winter Games.
The impact of the assimilation of this data into a numerical weather prediction model and its impact for winter storm prediction is in general an interesting topic. However, from my perspective the publication of the present manuscript can only be considered after substantial revision.

**Major comments:**
- The conclusions drawn from the two experiments are partly self-evident and don't provide much useful insight. It is shown that the assimilation draws the analysis closer to the assimilated observations, however, unless there is a bug in the system, this is to be expected. Moreover the authors show and discuss over several paragraphs that the effect of the assimilation are lost quickly during the forecast by the atmospheric dynamics. This is also a normal effect in the dynamic atmosphere, so for the reader it is not so much of interest **that** it happens, but rather **how long** the desirable positive impact lasts. Also the conclusion that A-B (which is actually the increment added in the assimilation step, which is not mentioned in the manuscript) has the same pattern with respect to positive and negative values as O-B is self evident and can be inferred directly from the equations of the data assimilation algorithms.
- I would suggest to rewrite and restructure the manuscript answering the following questions more clearly:
    - What data is assimilated? Describe the GPM mission (I think it is not even mentioned that this is satellite-based...)  and not only the retrival of the variables
    - What is the effect on the **analysis**? Verification against independent variables would be desirable
    - What is the effect on the **forecast** (on different variables such as precipitation, temperature...), and how long does it last?
- Motivation: For an improved prediction of snowstorms probably upper air observations would be more useful than these surface observations. This does not mean it is not interesting to exploit this surface data for data assimilation, I just wonder whether the motivation could be formulated slightly broader, e.g. to find out which variables can be improved during these two case studies, and which not.
- Figures: The manuscript contains many too many figures, and many of them do not contain conclusive information, thus can be omitted without loss of information
    - The scatterplots in Figure 5 could be omitted
    - Figure 7 and 8: Can be omitted, it only proves that the atmosphere is a dynamic (chaotic) system
    - Figure 9 can be omitted, Figure 10 is much more informative
    - Figure 11: a RMSE value would probably give more information than this Figure
    - Table 2: It would be much more illustrative to have some visualization here… Time-series or bar plots for example

**Minor comments**

- L90: This sentence is self-evident. Should be omitted.
- L103: What kind of observations?
- L109: ocean surface meteorology conditions is too unspecific here
- L128-130: What kind of radars are the KMA radars, and what exactly is PIP
- L156-158: This last sentence of the paragraph is not clear to me
- Page 7: Which model domain and which resolution was actually used for the data assimilation experiments? Was the data assimilated into all three model runs?
- L 223-225: Is xb the same as xguess?
- L27: What is pseudo relative humidity?
- L237: What kind are the satellite wind data? Atmospheric motion vectors? Or Scatterometer data?
- L251: What do you mean the data assimilation results are compared with WRF simulations? Don't you actually compare analyses and forecasts with observations?
- L269: Isn't actually the humidiy of the model higher than that of the observation, if you get a negative observation minus background?
- L291: A-B is actually the increment that is added to B in the assimilation step.

---

## Author Comment (AC2)

**Title:** Assimilation of GPM-retrieved Ocean Surface Meteorology Data for Two Snowstorm Events during ICE-POP 2018

**By:** Xuanli Li, Jason B. Roberts, Jayanthi Srikishen, Jonathan L. Case, Walter A. Petersen, GyuWon Lee, and Christopher R. Hain

**Submitted to:** *Geoscientific Model Development*

*We would like to thank the reviewers and the editor for their thorough analysis of our manuscript and for suggesting changes that will help our paper to have a better quality. We have reproduced the reviewer's comments below — our responses appear in* blue. *The line number in this response refers to the manuscript without track changes.*

**Reviewer #1 comments:**

General comments

This manuscript intends to investigate the impact of the assimilation of the Global Precipitation Measurement (GPM) retrieved ocean surface observations on two winter storms during 2018 using GSI and WRF model. The discussions primarily focus on the analysis impacts in the observation space, and the precipitation predictions for each case. This study has potentials in improving our understanding of how to optimally utilize the GPM retrievals. But the unclear motivation and lack of in-depth discussions make the paper reading frustrating. I would not recommend publication of this work until substantial revisions have been made. The detailed comments are listed below.

*We have enhanced introduction section on the motivation (please see the last paragraph in Page 4 Line 113-134) and the data section (please see Line 153-166). We have made substantial changes throughout the result section and the conclusion section. We have also added one more figure to discuss how the data assimilation modified the moisture convergence and moisture transport, hence influenced the precipitation production (please see Line 396-408 in Page 13).*

*For the motivation of this research, as we stated in the introduction section, is that there has been very little effort to assimilate these types of observations in modelling systems. It has not been an emphasis in the satellite-based flux community methods focused on estimating the near-surface meteorology have been concentrated. So, this is a unique opportunity to test the assimilation of such a dataset. For this particular paper, the focus is on establishing the ability to model the systems with WRF, implementing a strategy to assimilate the widespread surface observations (e.g. the technical implementation, data thinning, etc.), and doing a characterization. We think that demonstrating the ability of the data to impact the model fields and understanding how those impacts were made are prerequisites to understanding whether or not to pursue more sophisticate process-based diagnostics (this will be the focus of a follow-on research).*

**Major comments/concerns:**

One major concern I have about this manuscript is the unclear goal. If this study is focusing on the assimilation of GPM retrievals as suggested by the title, then the introduction should provide more information about the GPM retrievals, such as how it was (or was not) previously applied/researched in the field of snowstorm predictions (or other similar fields), and what's the novelty of this current study (not just the technical specifics in section 2.1). Also, the results section should contain more discussions on how the corrections from this dataset are physically benefiting the storm predictions, not just showing there is a difference and the model is improved. This is a case study after all. Please consider re-organize the manuscript/title to better represent your point and outstand your novelty.

*We have enhanced the introduction section by adding the following discussion on the GPM retrieved data, as well as the motivation and novelty of the current research: "In the satellite-based surface flux community (e.g. see Curry et al., 2004), significant efforts have been undertaken to estimate the near-surface meteorology from passive microwave observations to support the development of turbulent flux estimates from space. In particular, efforts have been made to estimate 2m air temperature and humidity (e.g., Jackson et al. 2006, Roberts et al. 2010, and Tomita et al., 2018) to complement long-standing wind speed estimates from microwave observation. However, the aforementioned efforts have almost explicitly focused on large-scale production of the fluxes for climatological analyses with long latencies. However, the surface retrieval products essentially provide similar measurements to those of buoys and generally with accurate performance. There is a long heritage of assimilating ocean surface buoy measurements within a data assimilation framework, but there has been little effort focused on assimilating the surface retrievals. This is part due to a lack of a real-time availability of these estimates and partly due to the focus on radiance-based assimilation system. The latter are not particularly tuned for leveraging lower-layer information in microwave observations as the stand-alone efforts originating from the satellite-derived flux community. The ICE-POP 2018 campaign provided a unique opportunity with near-real time passive microwave estimates of surface meteorology and a heavily observed regional environment to test the potential impact of assimilating wide-spread observations of near-surface meteorology. In this research, we explored the assimilation of this dataset using case studies with two snowstorm events occurred during the ICE-POP 2018 period. The objectives of the current research are to characterize the forecast ability of snowstorm events over complex terrain with the WRF model and further to develop and evaluate an approach to assimilate the passive microwave derived surface meteorology. Our focus herein emphasizes the large-scale impacts of assimilation of the surface meteorology on the corresponding model fields and downstream forecast accuracy."*

*A new figure (Figure. 10, and discussions in Line 396-408 in Page 13) is also added to discuss how the data assimilation influenced the low-level moisture convergence and moisture transport and hence benefit the precipitation forecast of the snowstorm. More details on the physical*

*processes (e.g. ocean evaporation, water and energy budget analyses, etc.) will be investigated in follow-on researches.*

The results section spends large portions of contents on how the background is modified towards the observations for each snowstorm event. But the discussions on either A-B or O-B can just tell us that the DA is working properly since the verification is dependent. Any successful DA should make the analysis closer to observations than the background, this is from the mathematical nature of the minimization of the cost function. As indicated between L126-132, there should be plenty of other observations (e.g. D3R) available in addition to the retrievals. Why not use those independent observations to verify your analysis? It will objectively measure if the DA is really improving the analysis.

*It is true that successful DA is meant to make the analysis closer to the observation than the background, but the data impact can vary, and sometimes it is case dependent. Therefore, it is important to know how close the analysis is to the observation. For this purpose, we compared A-B and O-B in Fig. 6 and Fig. 12 to examine how much influence the data assimilation brought to the model fields, temperature, specific humidity, and wind speed, and how close the analysis fields are to the observations. It is shown in Figs. 6 and 12 that the data impact does follow the pattern in O-B, but it varies in different events, even with the same DA procedure and parameters.*

*Yes, there are plenty of data collected from the ICE-POP 2018, as we mentioned in the last paragraph on Page 4. Some of the data are more readily to be used for model verification than the others. In the past research, we have examined the hourly precipitation rate retrieved from multiple ground-based radars (including D3R). We've also compared it with IMERG precipitation and found inconsistency between them due to the complicated nature of solid precipitation particles and the operational mode of the radars. Therefore, in this study, we choose to use the South Korean Surface Analysis dataset for model verification. This dataset is in-situ data based on observations from the dense Automatic Weather Station network and have a better accuracy in snowfall measurement. This 1-km resolution dataset is particularly produced for ICE-POP 2018. In the previous manuscript, we forgot to stress this point in the Data and Methods section. We've updated this information in our new manuscript (Line 154-160).*

**Minor but still important comments:**

L220: The cost function is not just to measure the difference between the model and observations (otherwise it should be just yobs-Hxb). It is the sum of weighted (B and R) differences between

the analysis estimation xa and model background xb (Jb), and between xa and observations y (Jo). The goal of DA is to find an optimal estimation xa that minimizes this J.

*Thank you for correcting us, we have revised this sentence in Line 252. Equation 1 itself explains the cost function very clear that it is the differences between the analysis and model background weighted by background error plus the differences between the observation and analysis weighted by observation error.*

L225: xguess is xb, please use just one subscription for consistency. This y term is also commonly known as innovation in recent DA literatures.

*Xguess is now updated as Xb and the y term is now defined as the observation innovation (please see Line 255).*

L240: This comment may be trivial, but can you provide more details on how you perform the cycling? There are two start mode in WRF, restart and cold start. In the restart mode, WRF is able to continue the model integration without interruption by using both the analysis at the current time step and the tendencies from the previous time step. However, our current DA methods usually only updates the analysis, not the tendencies. I'm wondering if you saw any discontinuity issue if you are using this mode. On the other hand, if you are using cold start mode (restart=.false. by default, which only uses the analysis at the current step), it will interrupt previous integrations and perform differently with your control experiments even without DA. You have to let the control exp stop at the same time for consistency.

*The cycled data assimilation is conducted in the following steps, taking the March case as an example:*

*For the experiment DA_Mar: The WRF model simulation started at 00 UTC 7 March. At 06 UTC 7 March, the 6-h WRF forecast was used as the background field and prepbufr data was assimilated. The boundary condition file was updated and the data assimilation analysis was used as the input file for the new WRF simulation. The WRF model was then ran for 3 hours. At 09 UTC, the 3-h WRF forecast file was used as the background field and the GPM retrieved meteorology data was assimilated. The data assimilation analysis was used as the input file for another new round of WRF simulation. The above-mentioned steps were repeated for each data assimilation cycle.*

*The GSI data assimilation only updates the model control variables, not the tendency terms. Therefore, the tendency terms from the previous time step may not be used after data assimilation. We checked the model fields, wind, pressure, etc, and did not find any apparent discontinuity issues in the model runs. The assimilation of the GPM-retrieved data might have interrupted the integration, but the same interruptions happened in the CTRL_MAR when the prepbufr data was assimilated every 6-h. Therefore, we think DA_MAR vs. CTRL_MAR, and DA_FEB vs. CTRL_FEB are fair comparisons.*

L261: Current Fig. 5 may not be the best way to show those info. I don't think it's possible to retrieve the percentile info directly from the figure.

*This sentence has been revised as: "The 25th (and 75th) percentile values of the observed surface temperature, specific humidity and wind speed are 4.96 (and 16.71) °C, 4.06 (and 8.70 g kg-1), and 6.04 (and 10.74 m s-1), respectively, which indicates that a larger part of the observational data located at the south part of Sea of Japan and western North Pacific Ocean than the north part." (Please see Line 299-302)*

L261-271: What do these statistics mean? Some of the o-b standard deviations are almost comparable or smaller than the corresponding observation errors (e.g. specific humidity). Are you suggesting your model background is very good already?

*The purpose of the statistics is to provide the overall distribution of the observational data and the distribution of the difference between the model background and the observations in temperature, specific humidity and wind speed fields. No, since the observational data was retrieved from satellite observations, we would like to use the observational errors recommended by the GSI system.*

L291-293: Maybe just show the RMSD of O-A and O-B to better present the point if you are not going to physically explain how those improvements can benefit the storm analysis or predictions?

*The RMSD has been calculated and discussed in Line 338-340 on Page 11.*

L299-301: I'm confused by the motivation of the discussion here. Differences should be there as long as you changed the initial analysis. Of cause the differences will spread out as the model tries to balance the changes. But without observations in those areas, how do we know if this "spread of information" is correct or not?

*Following Reviewer #2's suggestion, we have removed this figure (previous Fig. 7). But we pointed it out because "the spread of information" and accumulation of the impact were caused by the cycled data assimilation, and the spread of information was not evenly distributed over the domain, even though we don't have observational data (out of the ICE POP 2018 domain) to evaluate the correctness of the information.*

L308-315: What is your localization length scale? Are these reducing differences in the area of your observations and increasing differences in the remote area reasonable? What's the physical explanations or guesses?

*For these two events, we have created our own background error statistics files because an effective data assimilation could not be made with the default regional background error file in GSI. The regional background error matrix (including horizontal length-scales, vertical length-scales, regression coefficients, etc. of the control variables) was created with 1-month 24-h and*

*12-h forecasts with the same WRF domains as the case studies using the WRFDA gen_be package. This point has been added in the updated manuscript (Line 260-264 Page 9). Sensitivity tests were conducted with different horizontal scale parameters for reasonable result from the assimilation of the GPM retrieved surface data. The current values of horizontal scale are set as 0.375, 0.75, and 0.75 in the GSI namelist file. This set gives a reasonable distribution of data increment for surface temperature, relative humidity, and wind speed as seen in Figs. 6 and 12.*

*The change in differences should be the effect of model dynamic adjustment during WRF integration, not from the data assimilation.*

L324-329: How should we interpret this result from Fig. 8? Is it reasonable for a continuously cycled DA experiment to become more alike the NoDA experiment after several cycles? It appears to me that the impact of DA is fading after cycling. Is this true? Also, the increasing RMSD of wind speed at almost all levels from 21 UTC to 22 UTC (Fig. 8d) seems to be inconsistent with the reducing total RMSD in Fig. 8a. Why?

*Figure. 7 (previous Fig. 8) shows the RMSD in surface temperature, specific humidity, and wind speed between DA_Mar and CTRL_Mar from 21 UTC 7 March to 03 UTC 8 March. The GPM retrieved surface data was assimilated at 21 UTC 7 March, followed by 6-h WRF integration. Fig. 8 indicates that the significant impact was made in the analysis field (21 UTC), the data impact declined by 17%, 15%, and 4% on average in the first hour of model integration on surface temperature, specific humidity, and wind speed, respectively. In addition, the changes can be seen not only in surface and low levels, but also in mid to high levels. But as shown in Fig. 8a, a large part of the data impact (0.78°C in temperature difference, 0.29 g/kg in specific humidity difference, and 1.59 m/s in wind speed difference) still retained in the 6-h forecast (03 UTC). Therefore, Fig. 8 does not support the idea that the continuously cycled DA experiment becomes more alike the NODA experiment.*

*Thank you for catching the mistake in the wind speed plot. The legend in the wind speed plot was incorrect. The gray line should be WSPD1 and the blue line should be WSPD. We have corrected it in the updated figure.*

Fig. 9 -14: Can you use something like ETS score to quantitively verify the precipitation predictions? Also, in your discussions with the precipitation patterns, can you physically relate them to your previous analysis impacts? E.g. how does a warmer temperature in the analysis within certain area result in more precipitation predictions.

*We added the ETS information for Fig. 9 (please see Line 388-395 for the discussion). ETS would not be necessary for Figs. 11 and 14 since the PDF is already provided to evaluate the forecast accuracy. We have also added Fig. 10 to explain how data assimilation modified the physics fields and hence influence precipitation (please see Line 396 – 408 on Page 13). More*

*investigation on the specific physically fields and processes that data assimilation has influenced will be explained in follow-on researches.*

L371-380 and others: Why does the accuracy of this IMERG matter? How is this related to the focus of this paper? It reads to me that the discussions on the IMERG is out of no where. If you think this is a important part of your project and is somehow related to the focus of this paper, please add more information in the intro.

*Because the ICE-POP 2018 is also a part of the GPM ground validation effort in terms of solid precipitation, we think it might make some interesting points on the ability of IMERG data for snowstorms like the ones discussed in this paper. It provides a great opportunity to estimate the accuracy of GPM data for winter precipitation over complex terrain. We have added this point in the discussion section Line 435-438, and also in the conclusion section Line 541-544.*

L407-410: Why does the specific humidity increase while the innovations are mostly negative?

*Specific humidity varies with temperature, pressure and water vapor mixing ratio, and specific humidity is not a control variable in GSI. In GSI, the moisture control variable is pseudo relative humidity. For a more direct illustration of the data assimilation effect, we have replaced the specific humidity in Figs. 6 and 12 with relative humidity. From the new figures, you will find the A-B in relative humidity follows the general pattern of O-B.*

L444-445: What makes the Feb case less significant than the Mar case? Any hypothesis? It appears to me that the O-B differences in Feb case is larger than the Mar case (Fig. 6 v.s. Fig. 12). Could it be related to my previous comment on the specific humidity?

*Yes, the larger O-B may be a contributor to the less significant impact of the GPM retrieved data for the Feb case (we have added this point in the conclusion section, please see Line 532-534 on Page 17). Also, the different synoptic configurations and the undersampling of observational data are also important factors for that.*

---

## Author Comment (AC3)

**Title:** Assimilation of GPM-retrieved Ocean Surface Meteorology Data for Two Snowstorm Events during ICE-POP 2018

**By:** Xuanli Li, Jason B. Roberts, Jayanthi Srikishen, Jonathan L. Case, Walter A. Petersen, GyuWon Lee, and Christopher R. Hain

**Submitted to:** *Geoscientific Model Development*

*We would like to thank the reviewers and the editor for their thorough analysis of our manuscript and for suggesting changes that will help our paper to have a better quality. We have reproduced the reviewer's comments below — our responses appear in* blue. *The line number in this response refers to the manuscript without track changes.*

**Reviewer #2 comments:**

General assessment:

The manuscript describes the assimilation of GPM-retrieved near-surface variables into the WRF model for two snow-storm case studies during the ICE-POP 2018 research and forecast programs during the Olympic and Paralympic Winter Games. The impact of the assimilation of this data into a numerical weather prediction model and its impact for winter storm prediction is in general an interesting topic. However, from my perspective the publication of the present manuscript can only be considered after substantial revision.

*We have enhanced introduction section (please see the last paragraph in Page 4 Line 113-134) and the data section (please see Line 153-166) in the updated manuscript. We have made substantial changes throughout the result section and the conclusion section. We have also added one more figure to discuss how the data assimilation modified the moisture convergence and moisture transport, hence influenced the precipitation production (please see Line 396-408 in Page 13).*

Major comments:

• The conclusions drawn from the two experiments are partly self-evident and don't provide much useful insight. It is shown that the assimilation draws the analysis closer to the assimilated observations, however, unless there is a bug in the system, this is to be expected. Moreover the authors show and discuss over several paragraphs that the effect of the assimilation are lost quickly during the forecast by the atmospheric dynamics. This is also a normal effect in the dynamic atmosphere, so for the reader it is not so much of interest that it happens, but rather how long the desirable positive impact lasts. Also the conclusion that A-B (which is actually the increment added in the assimilation step, which is not mentioned in the manuscript) has the same pattern with respect to positive and negative values as O-B is self evident and can be inferred directly from the equations of the data assimilation algorithms.

*Yes, it is true that a successful data assimilation will drive the analysis closer to the observation. But it is also true that a reasonable analysis cannot be created without fine tune the data assimilation system and the parameters. Also, the data impact on different weather systems can be case dependent (as confirmed by the two storms in this paper). Therefore, it is necessary to conduct these experiments to demonstrate how the satellite retrieved observation influences the initial condition and the forecast fields. Actually, there has been very little effort to assimilate these surface flux retrieved meteorology observations in modelling systems. It has not been an emphasis in the satellite-based flux community. Therefore, this is unique to test the assimilation of this dataset. In this paper, the focus is to demonstrate the data assimilation, investigate how long the data impact will last and by how much the data assimilation will influence forecast. These will be very useful in designing the data assimilation cycles in order to retain the information from the observations. A-B is the increment, this has been added in Line 318.*

• I would suggest to rewrite and restructure the manuscript answering the following questions more clearly:

◦ What data is assimilated? Describe the GPM mission (I think it is not even mentioned that this is satellite-based...) and not only the retrieval of the variables

*In this study, we assimilated the ocean surface meteorology data 2-m temperature, 2-m specific humidity, and 10-m wind speed retrieved from GPM microwave observations. This point is mentioned in the introduction section Line 111, and in the data section Line 178 and the experiment section 2.2 Line 275. More discussion of this dataset has been added in Line 114-121. The description of GPM mission was added in the data section Line 161-166.*

◦ What is the effect on the analysis? Verification against independent variables would be desirable

*Figs. 6 and 12 shows the data assimilation increment on surface temperature, relative humidity and wind speed for March and February case, respectively. In the updated version, Root Mean Square Difference over the model domain was calculated for these variables too (Page 11 Line 337-341). There are not much other observations over the ocean could be used for direct verification, except some buoy data, which has already been used for validation of the GPM retrieved ocean surface observation.*

◦ What is the effect on the forecast (on different variables such as precipitation, temperature...), and how long does it last?

*Fig. 7 shows the impact of the data on the analysis and the RMSD on 1-6h surface temperature, specific humidity, and wind speed forecast. The impact has a large decline in the first hour of integration for temperature, specific humidity, and wind speed, but a large portion of the impact could still be seen in the 6-h forecast. Figs. 8 and 13 shows the impact of the data on*

*temperature forecast. Figs. 9, 10, 13 shows the impact of the data on 1-h accumulated and 24-h accumulated precipitation fields.*

• Motivation: For an improved prediction of snowstorms probably upper air observations would be more useful than these surface observations. This does not mean it is not interesting to exploit this surface data for data assimilation, I just wonder whether the motivation could be formulated slightly broader, e.g. to find out which variables can be improved during these two case studies, and which not.

*We have enhanced the introduction section by adding the following discussions on the motivation of the current research "There is a long heritage of assimilating ocean surface buoy measurements within a data assimilation framework, but there has been little effort focused on assimilating the surface retrievals. This is part due to a lack of a real-time availability of these estimates and partly due to the focus on radiance-based assimilation system. The latter are not particularly tuned for leveraging lower-layer information in microwave observations as the stand-alone efforts originating from the satellite-derived flux community. The ICE-POP 2018 campaign provided a unique opportunity with near-real time passive microwave estimates of surface meteorology and a heavily observed regional environment to test the potential impact of assimilating wide-spread observations of near-surface meteorology." (please see Page 4 Line 120-127).*

• Figures: The manuscript contains many too many figures, and many of them do not contain conclusive information, thus can be omitted without loss of information

*We have removed the previous Fig. 7, but do think the other 14 figures are necessary in order to effectively convey our conclusions from the two case studies. These figures show how and by how much the assimilation of this particular dataset influences the model initial condition and forecast for snowstorm events. These figures also show how consistent and different the data assimilation impacts are for the two different storms.*

◦ The scatterplots in Figure 5 could be omitted

*Figure 5 gives us the general statistics of the observational data and the distribution of O-B that have been assimilated into the model. We think it's important to keep Fig. 5 in the manuscript.*

◦ Figure 7 and 8: Can be omitted, it only proves that the atmosphere is a dynamic (chaotic) system

*Figure. 7 has been removed. Figure.7 (previous Figure. 8) shows how much data impact was produced in the analysis field, and how long and how much of the data impact would last in the forecast. We think Fig. 8 should be kept in the manuscript.*

◦ Figure 9 can be omitted, Figure 10 is much more informative

*We agree that, with the 24-h accumulated precipitation, Fig. 11 (previous Fig. 10) has more information than Fig. 9. But we prefer to keep Fig. 9 in the manuscript, we also added a new Fig. 10 for moisture convergence and moisture transport to explain the physical reason for the difference between Fig. 9b and 9c.*

◦ Figure 11: a RMSE value would probably give more information than this Figure

*We have provided the RMSE value in the updated manuscript (please see Page 12 Line 376-377).*

◦ Table 2: It would be much more illustrative to have some visualization here… Time-series or bar plots for example.

*It would be good to keep it as Table 2 considering that we already have 14 figures.*

**Minor comments**

• L90: This sentence is self-evident. Should be omitted.

*This sentence has been removed.*

• L103: What kind of observations?

*It's dropsonde data. We have added this point in the manuscript (please see Line 103).*

• L109: ocean surface meteorology conditions is too unspecific here

*We have revised this sentence as "In support of the International Collaborative Experiments for PyeongChang 2018 Olympic and Paralympic Winter Games (ICE-POP 2018) field campaign, ocean surface meteorology conditions (2-m temperature, 2-m specific humidity, and 10-m wind speed) were retrieved from the Global Precipitation Measurement (GPM) microwave observations from January to March 2018" (please see Line 111).*

• L128-130: What kind of radars are the KMA radars, and what exactly is PIP

*The KMA radars are S-band Doppler radars. This information has been added in the manuscript Line 149. The Precipitation Imaging Packages (PIP) is a video disdrometer made up of a single high-speed camera, continuously recording at 380 frames per second, and a halogen lamp to backlight the precipitation particles.*

• L156-158: This last sentence of the paragraph is not clear to me

*This sentence has been removed from the manuscript.*

• Page 7: Which model domain and which resolution was actually used for the data assimilation experiments? Was the data assimilated into all three model runs?

*The data were assimilated into all 3 domains.*

• L223-225: Is xb the same as xguess?

*Yes, we have modified Xguess as Xb for consistency.*

• L27: What is pseudo relative humidity?

*Pseudo relative humidity (PRH) is defined by the following equation:*

$$PRH = \frac{r}{r_{sb}(T_b, P)}$$

*where r is the mixing ratio, $r_{sb}$ is the mixing ratio of a volume of air that is saturated with water vapor from background, which is affected by temperature from background $T_b$ and atmospheric pressure P.*

• L237: What kind are the satellite wind data? Atmospheric motion vectors? Or Scatterometer data?

*It includes satellite derived wind reports from GOES satellites and scatterometer wind vector (according to https://www.emc.ncep.noaa.gov/mmb/data_processing/prepbufr.doc/table_1.htm and https://www.emc.ncep.noaa.gov/mmb/data_processing/Satellite_Historical_Documentation.htm#Sec.%20II).*

• L251: What do you mean the data assimilation results are compared with WRF simulations? Don't you actually compare analyses and forecasts with observations?

*Thank you so much for catching that. This sentence has now been revised as "In this section, WRF simulations with and without the assimilation of the GPM-derived ocean surface meteorology data were compared with the observations collected for the March 7-8 and February 27-28 snowstorm cases." (Please see Page 9 Line 287-289).*

• L269: Isn't actually the humidity of the model higher than that of the observation, if you get a negative observation minus background?

*Correct, the model background has a higher specific humidity than the observation. This has been corrected in the manuscript in Line 308.*

• L291: A-B is actually the increment that is added to B in the assimilation step.

*Correct, A – B is the increment in the assimilation step (added in Line 318. This sentence has been revised (Please see Page 11 Line 335-337).*

---

## Author Response (AR2)

**Response to Reviewers' Comments on the original submission of**

Assimilation of GPM-retrieved Ocean Surface Meteorology Data for Two Snowstorm Events during ICE-POP 2018

*We would like to thank the reviewers and the editor for their thorough analysis of our manuscript and for suggesting changes that will help our paper to have a better quality. We have reproduced the reviewer's comments below — our responses appear in blue. The page and line numbers refer to the trackchanged manuscript.*

**Comments from Reviewer #1:**

General comments
This is the second-round review of the manuscript. The authors have made efforts to address my previous concerns. Nevertheless, I am not satisfied with the revisions made so far. The revised manuscript is still lacking a clear goal, and too many unnecessary details are in fact very distractive. I still do not think this is a paper ready for publication in its current form. The detailed comments are listed below.

Thank you so much for the comments, and thank you so much for recognizing our effort in revising this paper. We have revised the objectives in the introduction section and the conclusion section. Please see Page 4-5 Line 128-132 and Page 17 Line 540-542.

Since the current paper is particularly written for the special issue "*Winter weather research in complex terrain during ICE-POP 2018 (International Collaborative Experiments for PyeongChang 2018 Olympic and Paralympic winter games)*" of GMD and the main purposes of the special issues are (https://gmd.copernicus.org/articles/special_issue10_1112.html):
"1) to document the scientific findings on the winter weather during the forecast demonstration project
 2) to share scientific knowledge on processes of winter weathers that have been investigated with unprecedented dense observational networks,
 3) to share current status and improved knowledge of forecasting of winter weathers, and
 4) to document new retrieval and quality control methods of the operational and advanced instrument"
In this paper, we have introduced the surface meteorology data that has been retrieved from the GMP microwave observation. This dataset was created particularly in support of the ICE-POP 2018 field campaign. The main goals of this paper are to introduce this ICE POP dataset, to explore the application of this dataset through data assimilation, to demonstrate the influence of this dataset, and to examine whether or not the assimilation of this dataset is able to improve the forecasts of two heavy snowstorm that has been occurred during ICE-POP 2018.

In the revised paper, we have removed unimportant details as the reviewer suggested, yet kept other necessary details in order to support our key result and conclusions.

**Major comments:**

1. In summary, the manuscript basically shows that there are significant changes made to the background and the forecasts in the temperature, precipitation and moisture flux divergence can be improved. Also, it shows that the changes made to the background can be different between Feb and

Mar cases. Speculations are due to the different kinematic and thermodynamic processes involved. As a case study, we cannot draw any statistically robust conclusions from the results. Yet the manuscript failed to provide any physical link on how the changes to the background can lead to the changes in the forecasts. The authors should try identifying at least one major contribution of the DA towards the better predictions. For example, which one of the T/Q/WSD fields is more important to the better predictions? Can the increased/reduced temperature analysis trigger more/less convection-induced precipitations in certain areas, which leads to better precipitation forecasts? What exact kinematic or thermodynamic process is the key for the Feb and Mar differences? etc… Sensitivity experiments are strongly recommended to explore the impact of DA for your single case study.

We agree with the reviewer that better understanding of the physics on how the data assimilation influenced the winter storm forecast is very important and of great interest. Therefore, in our previous manuscript, we have added Fig. 10 to illustrate how the data assimilation influenced the low level (925 hPa) moisture flux convergence and moisture transport. We have found that the overestimate of precipitation in CTRL_Mar was improved in DA_Mar because of the weaker low level moisture convergence and moisture flux.

A thorough understanding of the physical mechanisms of the winter precipitation is a hot topic, and it is also one of the main research goals of ICE POP 2018 project. However, it requires a great amount of research and cannot be covered completely in the current paper. Forecast of winter precipitation in South Korea is particularly challenging due to its unique geographic features and large-scale circulation. Orographic effect, synoptic circulation, and air-sea interaction can all have important roles and interact with each other. Sensitivity studies and in-depth investigations on different dynamic, thermodynamic, and microphysics factors and processes (both large scale and local effect) need to be taken place to examine how, by how much, and for how long these factors can influence the winter precipitation and the model forecast. Some to the publications in this special issue tried to tackle this topic. For example, Gehring et al. (2020) indicated that a warm conveyor belt (WCB) had significant impact on the microphysics processes for the heavy snow storm on 28 February 2018. The WCB had an important role in generating supercooled liquid water and enhancing aggregation which led to rapid precipitation growth. Kim et al. (2021) showed that wind pattern and topography affect the microphysical processes for snowstorms over northeast part of south Korea. Strong wind shear and turbulence were the cause of the intense riming process that dominates over the mountainous region. For the coastal region, aggregation becomes more important than riming in generating precipitation.

On our end, further investigation will be performed to better understand how data assimilation modified these factors and processes which lead to a better forecast. These research activities require a great amount of work that will be conducted in our future studies related to this project.

Reference:
Kim, K., W. Bang, E.-C., Chang, F. J. Tapiador, C.-L. Tsai, E. Jung, and G. Lee, 2021: Impact of wind pattern and complex topography on snow microphysics during ICE-POP 2018. Atmos. Chem. Physi. 21, 11955–11978, https://doi.org/10.5194/acp-2021-128

Gehring, J., A. Oertel, É. Vignon, N. Jullien, N. Besic, and A. Berne, 2020: Microphysics and dynamics of snowfall associated with a warm conveyor belt over Korea. Atmos. Chem. Phys., 20, 7373–7392, https://doi.org/10.5194/acp-20-7373-2020.

2. L313-341: I see the authors have reduced the corresponding discussions. Yet I still don't see the

value of this O-B vs A-B comparison. Yes, DA can perform differently, and increment/influence can be large/small depends on the methodology/configurations. But in this current paper, the authors never compared any DA schemes other than a single scheme in two cases. I don't get what to expect from these comparisons. Again, it is a common knowledge that a successful DA is supposed to drag the analysis towards observations. Then, what else can we get from here? Are the increments balanced? Can the corrections be validated from other independent obs? I think my point is that you can use a few words/O-B/O-A statistics to support the idea that DA is working properly, but no need to use the figure and too much paragraph. Otherwise, try dig something new/unexpected.

Thank you for the comments and thanks the reviewer to agree with us on some of our points. In the revised manuscript, we tried our best to shorten this paragraph. Since the dataset contains 3 types of observation, we used a few sentences to describe the impact of each data type. We do want to keep the RMSD numbers at the end of the paragraph because this was recommended by reviewer #1 in the previous round of review.

Since this paper focuses on case studies for assimilating a new dataset with high resolution regional simulation, it is always a good idea to check the background, the observation innovation (O – B) and the analysis increment (A – B) and see whether or not the data assimilation is effective (similar plots can be found in numerous past studies such as Xiao et al. 2007, Bi et al. 2011, Chen et al. 2021, etc). The RMSD numbers can provide a general sense of how the model fields respond to data assimilation. However, the plots of O – B and A – B provide the horizontal distribution of the obs innovation and analysis increment. By comparing them, it is verified that the influence of the data is producing a reasonable spatial distribution.

Reference:

Xiao, Q., Kuo, Y., Sun, J., Lee, W., Barker, D. M., & Lim, E. (2007). An Approach of Radar Reflectivity Data Assimilation and Its Assessment with the Inland QPF of Typhoon Rusa (2002) at Landfall, *Journal of Applied Meteorology and Climatology*, *46*(1), 14-22.

Bi, L., Jung, J. A., Morgan, M. C., & Le Marshall, J. F. (2011). Assessment of Assimilating ASCAT Surface Wind Retrievals in the NCEP Global Data Assimilation System, *Monthly Weather Review*, *139*(11), 3405-3421.

Chen, S., Shih, C., Huang, C., & Teng, W. (2021). An Impact Study of GNSS RO Data on the Prediction of Typhoon Nepartak (2016) Using a Multiresolution Global Model with 3D-Hybrid Data Assimilation, *Weather and Forecasting*, *36*(3), 957-977.

3. L342-359: I don't think this exp design is a good approach to show how long or how much the DA impact lasts. First, CTRL_Mar is not initialized from the same background as DA_Mar, so the differences are not purely from DA. Second, since you are not verified against truth/nature run, the differences are not informative. In an extreme example, if the whole system is shifted from south to north (say, closer to the truth), yet you can still get the same average profile, and the difference won't grow if the parallel system moves similarly. Current configuration only shows that the DA forecast is being balanced after roughly 4~6 hours. Instead, you should use the statistics from the RMSE of Fig. 8 (or some other observation verifications) to do the impact study.
PS: I also still don't understand why the authors want to use only the South Korean Surface Analysis dataset for verification. Isn't it better to use the datasets with more confidence (e.g. examined before) for verification purposes?

We have updated Fig. 7 for a different time. In the current manuscript, the computation was conducted for the forecast after the 1$^{st}$ data assimilation cycle (0600 UTC 7 March 2018). DA_Mar assimilated both GPM-retrieved data and conventional data at 0600 UTC 7 March 2018 and CTRL_Mar assimilated only conventional data at 0600 UTC 7 March 2018. CTRL_Mar and DA_Mar used the same background field, so they should be a fair comparison.

Thank you so much for the suggestion of using the RMSD between the observational data and DA_Mar as in Fig. 8 to examine the magnitude and lasting period for data impact. However, this may not be a plausible method. Firstly, unlike OSE or OSSE, a real case study like this does not possess a nature run which is error free and has the same variables with perfect (or seemingly perfect) data. Secondly, the nature run of OSE or OSSE is at the same horizontal and vertical grids as the sensitivity experiment. However, in our case, the South Korean Surface Analysis that we used only covers continental South Korea (as shown in Fig. 8a). And the observational data that we have assimilated are over the ocean (as shown in Fig. 1). The analysis fields shown in Figs. 6 and 12 clearly indicated that most impact was found over ocean near the observed location and very slight impact was found over Korean Peninsula (another reason for keeping Figs. 6 and 12 in the paper). Therefore, if we use observational data over continental South Korea to do the evaluation on data impact, we will miss the vast area over the ocean where the majority impact occurred. This will likely to hold for the forecast within the first few hours of integration. That will not be a fair evaluation.

In this study, we choose to use the South Korean Surface Analysis dataset for independent verification because this dataset is in-situ data based on observations from the enhanced Automatic Weather Station network during the ICE POP 2018, hence have a better accuracy in snowfall measurement than the remote sensing observations. The dataset has been verified against the AWS observations over South Korean. According to Ryu et al. (2021), the methodology for this dataset has been examined by many past studies (e. g., Ozturk et al. 2014, Guo et al. 2011, Li and Parker 2008) which showed that this method may surpass the traditional limits of sampling theory and recover missing spatial or temporal data. Recent researches (e. g., Demissie et al. 2021; Chen et al. 2021) also adopted and cited Ryu et al. (2019).

Reference:

Chen, H., S. Sheng, C.-Y. Xu, Z. Li, W. Zhang, S. Wang, and S. Guo, 2021: A spatiotemporal estimation method for hourly rainfall based on F-SVD in the recommender system. Environmental Modelling & Software, 144, 105148, https://doi.org/10.1016/j.envsoft.2021.105148.

Demissie, T. A., and C. H. Sime, 2021: Assessment of the performance of CORDEX regional climate models in simulating rainfall and air temperature over southwest Ethiopia, Heliyon, 7, e07791. https://doi.org/10.1016/j.heliyon.2021.e07791.

Guo, D., H. L. X. Qu, and Y. Yao, 2011: Sparsity-based spatial interpolation in wireless sensor networks. J. Sensor, 11, 2385–2407.

Li, Y.Y., and L. Parker, 2008: Classification with missing data in wireless sensor network. IEEE SoutheastCon 2008, pp. 533–538.

Ozturk, S., T.-Y. Yu, L. Ding, R.D. Palmer, N. A. Gasperoni, 2014: Application of Compressive Sensing to Refractivity. IEEE Trans. Geosci. Remote Sens. 52, 2799–1809.

Ryu, S., J. J. Song, Y. Kim, S.-H. Jung, Y. Do, and G. Lee, 2021: Spatial Interpolation of Gauge Measured Rainfall Using Compressed Sensing, Asia-Pacific J. Atmos. Sci., 57, 331–345. https://doi.org/10.1007/s13143-020-00200-7.

4. L360-377, L380-390 & others: Throughout the manuscript, there are too many tedious fact descriptions and number counting without clear goals, which make the readers lose focus. Unless there is any physically important point, the detail listing is very distractive and unnecessary. For example, what's the major point here other than DA_Mar has less RMSE? Why are those spots in Taebaek Mountains or Sobaek Mountains so important? Can they physically influence the storm predictions somehow? You should really connect the details to specific improvements to make a meaningful point, otherwise, just summarize them with one or two sentences.

We have revised the discussion in L360-377, L380-390, and other places as well. Since most of the readers of GMD may live in countries other than South Korea, the inclusion of necessary knowledge about the South Korean cities and geography might be very helpful for them, so we kept some details that are important to support our conclusions.

5. L435-449: I still do not think it is a good way to discuss the quality of GPM precipitation data here. It is off topic, and you should evaluate this irrelevant (to the goal of this manuscript) dataset in another paper with more thorough analyses.

We have removed the discussion on the GPM IMERG data, please see Page 14, 16, and 17. The plots related to GPM observation have also been removed from Figs. 8 and 10.

**Minor comments:**

1. L115: 2m -> 2-m.
Corrected.

2. L119: Please try to reduce the use of however.
This has been changed to "on the other hand".

3. L122: part -> partly
Corrected.

4. L301-302: How to deduce this implication from the figures or percentiles?
This was not deduced from the figure, this sentence has been revised.

5. L303-310: Did you assume the observations to be unbiased? And again, what do these numbers mean? Model is biased?
Yes, there was no bias correction adopted in the current data assimilation procedure. From the $O - B$ scatterplots, it is shown that the departures between observation and model field are skewed to different extent. In future studies, we will examine the data for a longer period of time. When we have more robust conclusions about the bias of this GPM-retrieved surface observation data, we may consider adding a bias correction procedure before or during the minimization for a better data assimilation result (this point has been mentioned in the conclusion section on Page 18 Line 569-576).

6. L338-341: Directly calculate O-A and compare it with O-B will be more straightforward than A-B.
We would prefer to keep the A – B because it is an import data assimilation term, the so-called analysis increment.

**Topical Editor decision: Reconsider after major revisions**
by Chiel van Heerwaarden

**Comments to the author**:
While the paper has improved, the reviewer is not happy with the revisions. I have to agree with the reviewer that the paper still needs improvement in clarity and focus. I am willing to provide you this opportunity and advise you to take into account the suggestions of the reviewer in detail. Most important: please set the goal of the paper clearly, and try to remove irrelevant information.

Thank you so much for the comments and thank you so much for providing us the opportunity. In the updated manuscript, we made significant revisions on the research goal. Please see the introduction section (Page 4-5 Line 128-132) and the conclusion section (Page 17 Line 540-542).

We have also updated Figs. 8 and 10. In the new figures, we have removed the plots as well as the discussion related to GPM IMERG data quality (please see Pages 14, 16, and 17) as suggested by the reviewer.

We have updated Fig. 7 with the plots for the first data assimilation cycle to address the reviewer's concern.

We have also revised the result section from Page 10 to 17 to make the manuscript more concise.

---

## Author Response (AR3)

**Response to Reviewers' Comments on the 3rd Version of Manuscript of**

Assimilation of GPM-retrieved Ocean Surface Meteorology Data for Two Snowstorm Events during ICE-POP 2018

*We would like to thank the reviewer and the editor for their thorough analysis of our manuscript and for suggesting changes that will help our paper to have a better quality. We have reproduced the reviewer's comments below — our responses appear in blue. The page and line numbers refer to the trackchanged manuscript.*

**Comments from Reviewer #1:**

Re-Review of
Assimilation of GPM-retrieved Ocean Surface Meteorology Data for Two Snowstorm Events during ICE-POP 2018
by
Li et al.

**General comments**
This is the third version of the manuscript. The authors have changed the focus of the manuscript and further removed some unnecessary discussions. The manuscript is now more focused and refined. I appreciate the efforts of the authors. Although I personally feel like the content is a little too thin (more in-depth diagnostics are recommended), I believe the manuscript is now closer to publication with a few more comments.

**Comments:**
1. L309-334: I agree with the authors that "Since this paper focuses on case studies for assimilating a new dataset with high resolution regional simulation, it is always a good idea to check the background, the observation innovation (O – B) and the analysis increment (A – B) and see whether or not the data assimilation is effective (similar plots can be found in numerous past studies such as Xiao et al. 2007, Bi et al. 2011, Chen et al. 2021, etc).". However, as far as I can tell from the three citations and lots of other studies, a reasonable check for the DA effectiveness usually starts with O-B vs O-A (e.g. Fig. 6 in Xiao et al. 2007; Fig. 1& 4 in Bi et al. 2011; Fig. 4 in Chen et al. 2021), not O-B vs A-B. One simplified example is A=3, O=2, B=1, O-B=1 & A-B=2. You'll see both red in Figure 6, yet you cannot say the overcorrection is an effective DA. It is not wrong to use A-B, just less straightforward. Also, why does A-B look much smoother than O-B? Are you plotting one in observation space, and the other one in model space? If so, please use a consistent approach.

Thank you so much for the comment. We agree with the reviewer that either O-A (as shown in the literatures mentioned above in the reviewer's comment) or A-B (e.g., Benjamin et al. 2004; Bölöni and Horvath 2010; Piccolo and Cullen 2016, to name a few) can be used to verify the data assimilation effect. We prefer to use A-B simple because A-B (the so-called analysis increment) provides the readers a direct vision on where and by how much the data assimilation corrects the model guess fields. In the example that the reviewer provided, A=3, O=2, B=1. Since the reviewer mentioned the color red, we think the reviewer was talking about the temperature field. Using these numbers, we will have O-B=1 and O-A=-1, meaning the analysis

overpredicts temperature. At the same time, with O-B=1 and A-B=2, we can reach the same conclusion that the analysis has an overprediction. This can be seen in Fig. 6. In Fig. 6b, we will see the color for O-B is golden (the golden color is for the value ≥ 1.0 and < 2.0). In Fig. 6c, we will see the color for O-A is orange (the orange color is for the value ≥ 2.0 and < 3.0). This tells us that the analysis has overpredicted temperature.

No, we did not smooth any of the fields and both plots are in the observation space. During 3dvar, the impact of the each obs data was distributed over an influence area which includes the observation location and other grid points (in both horizontal and vertical direction), that's why we see smoother fields in analysis.

**Reference:**

Benjamin, S. G., Dévényi, D., Weygandt, S. S., Brundage, K. J., Brown, J. M., Grell, G. A., Kim, D., Schwartz, B. E., Smirnova, T. G., Smith, T. L., & Manikin, G. S. (2004). An Hourly Assimilation–Forecast Cycle: The RUC, *Monthly Weather Review*, *132*(2), 495-518. https://journals.ametsoc.org/view/journals/mwre/132/2/1520-0493_2004_132_0495_ahactr_2.0.co_2.xml

Bölöni, G., and Horvath, K. (2010). Diagnosis and tuning of background error statistics in a variational data assimilation system. IDŐJÁRÁS Quarterly Journal of the Hungarian Meteorological Service, 114, 1 – 19.

Piccolo, C., and Cullen, M. (2016). Ensemble Data Assimilation Using a Unified Representation of Model Error, *Monthly Weather Review*, *144*(1), 213-224. Retrieved Jun 6, 2022, from https://journals.ametsoc.org/view/journals/mwre/144/1/mwr-d-15-0270.1.xml

2. Fig. 7 & L335-375: I'm still not fond of this comparison. The evolution of the difference between CTL & DA can tell us very limited information. It can only show us that the model is being stabilized after a while, but it cannot tell us if the changes are good or bad. If the authors cannot find appropriate observations, ECMWF analysis or FNL re-analysis can be other potential sources for verifications and are commonly used in early studies.

Thank you so much for the suggestion, but we don't think the global analysis or reanalysis like ECMWF or FNL would be good candidates for verification for this particular study. First, these analyses have a much coarser resolution (30 km resolution for ERA5 or 1 deg to 0.25 deg resolution FNL analysis) when compared to our model simulations (9 km + 3 km + 1 km). With the resolution of 30 km or .25 deg, these analyses would not be able to describe the local features over the complex terrain in Korean Peninsula, but these are important features to look at for the cases (e.g., Figs 8-11, 13-14) that we examined in this paper. Second, many literatures indicated that the global analyses and reanalyses have problems in providing accurate description for winter storms (e.g., Hamill et al. 2013 for case studies and Feser et al. 2021 for statistical analysis). Before the control run, we have looked at the FNL analysis and also conducted test runs WRF forecast using the FNL analysis as the initial condition and the result supported the statement. This served the motivation for our data assimilation experiments which incorporated additional (prepbufr and the satellite-retrieved) datasets.

**Reference:**

Feser, F., Krueger, O., Woth, K., & van Garderen, L. (2021). North Atlantic Winter Storm Activity in Modern Reanalyses and Pressure-Based Observations, *Journal of Climate*, *34*(7), 2411-2428. https://journals.ametsoc.org/view/journals/clim/34/7/JCLI-D-20-0529.1.xml

Hamill, T. M., Yang, F., Cardinali, C., & Majumdar, S. J. (2013). Impact of Targeted Winter Storm Reconnaissance Dropwindsonde Data on Midlatitude Numerical Weather Predictions, *Monthly Weather Review*, *141*(6), 2058-2065. https://journals.ametsoc.org/view/journals/mwre/141/6/mwr-d-12-00309.1.xml

3. Although the authors have improved the manuscript, there are still some irrelevant number listings. For example,
(a) L297: What is the point of the median values of the observations? They are time-/case-sensitive values. What can the readers learn from them?

The median values of the observations have been removed from the content. Please see Page 10 Line 296-298.

(b) L300-308: Although the skewness can be calculated in a variety of ways, I don't believe only the mean and standard deviation can be enough (at least a median value of those departures is needed depending on the calculation, but then the earlier median values can be misleading). Again, what is the point of those values here? It might be better to use skewness values here instead, or you can just use the figure to illustrate the points.

We have revised the manuscript. The information about the mean and standard deviation was removed. The sknewness values for the deference fields were added. Please see Page 10 Line 302-309.